# Oxytocin restores context-specific hyperaltruistic preference

Hong Zhang[1†], Yinmei Ni[1*†], Jian Li[1,2*]

[1]School of Psychological and Cognitive Sciences and Beijing Key Laboratory of Behavior and Mental Health, Peking University, Beijing, China; [2]IDG/McGovern Institute for Brain Research, Peking University, Beijing, China

## eLife Assessment

This revised paper provides **valuable** findings that altruistic tendency during moral decision-making is gain/loss context-dependent and oxytocin can restore the absence of altruistic choices in the loss domain. The methods and analyses are **solid**, yet the study could still benefit from better overall framing and more clarity and precision in the definition of key constructs, as pointed out by reviewers. If these concerns are addressed, this study would be of interest to social scientists and neuroscientists who work on moral decision-making and oxytocin.

**\*For correspondence:**
niyinmei@pku.edu.cn (YN);
leekin@gmail.com (JL)

[†]These authors contributed equally to this work

**Abstract** Recent advances in moral decision-making research show people are hyperaltruistic by being more willing to sacrifice monetary gains to spare others from suffering than to spare themselves. Yet other studies indicate an opposite egoistic bias: subjects are less willing to harm themselves for others' benefits than for their own. These results underscore the complexities of moral decisions and demand a mechanistic explanation for hyperaltruistic preferences. We investigated hyperaltruism using trade-off choices combining monetary gains and painful electric shocks and choices combining monetary losses and shocks. Study 1 revealed that switching the decision context from gains to losses effectively eliminated the hyperaltruistic preference, accompanied by the altered relationship between subjects' instrumental harm (IH) trait attitudes and relative pain sensitivities. In the pre-registered study 2, we found that oxytocin, a neuropeptide linked to parochial altruism, restored the context-dependent hyperaltruistic preference. Furthermore, oxytocin increased the degree to which subjects framed the task as harming others, which mediated the correlation between IH and relative pain sensitivities. Thus, the loss decision context and oxytocin diminished and restored the mediation effect of subjective harm framing, respectively. Our results help elucidate the psychological processes underpinning the contextual specificity of hyperaltruism and carry implications in promoting prosocial interactions.

## Introduction

The Chinese proverb "A virtuous man acquires wealth in an upright and just way" stresses the universal moral code of refraining from harming others for personal gain (*Kahane et al., 2018*). Disregard for the suffering of others is often associated with aggressive and antisocial tendencies in psychopathy (*Bartels and Pizarro, 2011*; *Glenn et al., 2010*). Recent studies further developed the moral decision theory and suggested that people are willing to pay more to reduce other's pain than their own pain, yielding a 'hyperaltruistic' preference in moral dilemmas (*Crockett et al., 2014*; *Crockett et al., 2015*; *Crockett et al., 2017*; *Volz et al., 2017*). For example, it was shown that the hyperaltruistic preference modulated neural representations of the profit gained from harming others via the functional connectivity between the lateral prefrontal cortex, a brain area involved in moral norm violation,

and profit-sensitive brain regions such as the dorsal striatum (*Crockett et al., 2017*). However, moral norm is also context dependent: vandalism is clearly against social and moral norms yet vandalism for self-defense is more likely to be ethically and legally justified (the Doctrine of Necessity). Therefore, a crucial step is to understand the boundary conditions for hyperaltruism. We set out to address this question by examining how the moral perception of decision context affects people's hyperaltruistic preference (*Bustan et al., 2018*; *Dreber et al., 2013*; *Dungan et al., 2017*; *Li and Xu, 2024*; *Linde and Sonnemans, 2015*). More importantly, we also test whether the contextual effect of hyperaltruistic disposition is susceptible to oxytocin, a neuropeptide heavily implicated in social bonding and social cognition (*Bartz et al., 2011*; *Crespi, 2016*; *Young, 2015*).

Classic moral dilemmas often involve the trade-off between personal material well-being and the adherence to social norm (or moral principle) (*Greene et al., 2001*; *Greene et al., 2004*). Previous studies have shown that the moral preference can be highly context specific. For example, studies showed that people were more likely to engage in unethical behavior (lie or cheat) to avoid monetary loss than to secure monetary gain (*Kim and Guinote, 2022*; *Zhang et al., 2023*; *Steinel et al., 2022*). Other research, instead, found that it was more common for people to help others from losing money than to help them to gain money (*Everett et al., 2015*; *Li et al., 2020*). However, the boundary conditions for hyperaltruism remain elusive (*Crockett et al., 2014*; *Crockett et al., 2015*; *Crockett et al., 2017*; *Volz et al., 2017*). Hyperaltruistic disposition was typically measured in a money-pain trade-off task where individuals' monetary gain was pitted against the physical pain experienced by others and themselves. In turn, the hyperaltruistic preference was calculated by comparing the amounts of monetary gain subjects were willing to forgo to reduce others' pain relative to the reduction of their own pain. Therefore, it is likely that replacing monetary gain with monetary loss in the money-pain trade-off task might bias subjects' hyperaltruistic preference due to stronger sensitivity toward monetary loss (loss aversion) or highlighted vigilance in the face of potential loss (*Kahneman and Tversky, 1979*; *Tom et al., 2007*; *Liu et al., 2020*; *Usher and McClelland, 2004*; *Yechiam and Hochman, 2013*; *Pachur et al., 2018*). Indeed, elevated attention to losses can alternate subjects' sensitivities to the general reinforcement structure and have distinct effects on their arousal and performance (*Yechiam and Hochman, 2013*). Additionally, prospective losses can elicit negative emotions (*De Martino et al., 2010*; *Charpentier et al., 2016*), which may subsequently influence how individuals evaluate their own pain relative to others'. For example, studies have shown that negative mood can both increase pain sensitivities and decrease empathy for others, both of which yield opposing effects on hyperaltruistic preferences (*Becker et al., 2017*; *Carlino et al., 2014*; *Jiang et al., 2021*; *Pozo et al., 2023*; *Ma et al., 2012*). Finally, how the moral dilemma is framed can be a critical factor influencing moral decision-making (*McDonald et al., 2021*; *Cao et al., 2017*; *Evans and van Beest, 2017*). For instance, explicitly manipulating moral frames as 'harming others' (harm frame) or 'not helping others' (help frame) resulted in a significant social framing effect such that subjects showed stronger prosocial preference in the harm frame compared to the help frame (*Liu et al., 2020*; *Yang et al., 2022*). It is thus possible that people may implicitly form moral impressions (help or harm) based on the decision contexts and adjust their behavior accordingly. This way, the contextual effect on hyperaltruistic preference can be associated with individuals' internal framing of moral contexts.

Oxytocin, a neuropeptide synthesized in the hypothalamus, has been shown to modulate a variety of emotional, cognitive, and social behaviors through distributed receptors in various brain areas (*Petrovic et al., 2008*; *Meyer-Lindenberg et al., 2011*). Oxytocin has been shown to play a critical role in social interactions such as maternal attachment, pair bonding, consociate attachment, and aggression in a variety of animal models (*Kendrick, 2000*; *Young and Wang, 2004*). Humans are endowed with higher cognitive and affective capacities and exhibit far more complex social cognitive patterns (*Dunbar and Shultz, 2007*). It has been suggested that oxytocin increases prosocial behaviors such as trust and altruism by enabling individuals to overcome proximity avoidance, enhancing empathy for others' suffering, and reducing the processing of negative stimuli (fearful or angry faces) (*Zak et al., 2007*; *Barchi-Ferreira and Osório, 2021*; *Abu-Akel et al., 2015*; *Radke et al., 2013*; *Evans et al., 2010*; *Kirsch et al., 2005*; *Wu et al., 2020*; *Kapetaniou et al., 2021*). However, the evidence for whether and how oxytocin influences hyperaltruism remains scarce (*Zheng et al., 2024*). This may be partly because the prosocial effect of oxytocin seems to be both personality trait and decision context dependent (*Bartz et al., 2011*; *Han and Ma, 2015*; *Ma et al., 2016*; *Ma et al., 2015*; *Liu et al., 2019*). For example, recent studies have shown that oxytocin both promotes in-group

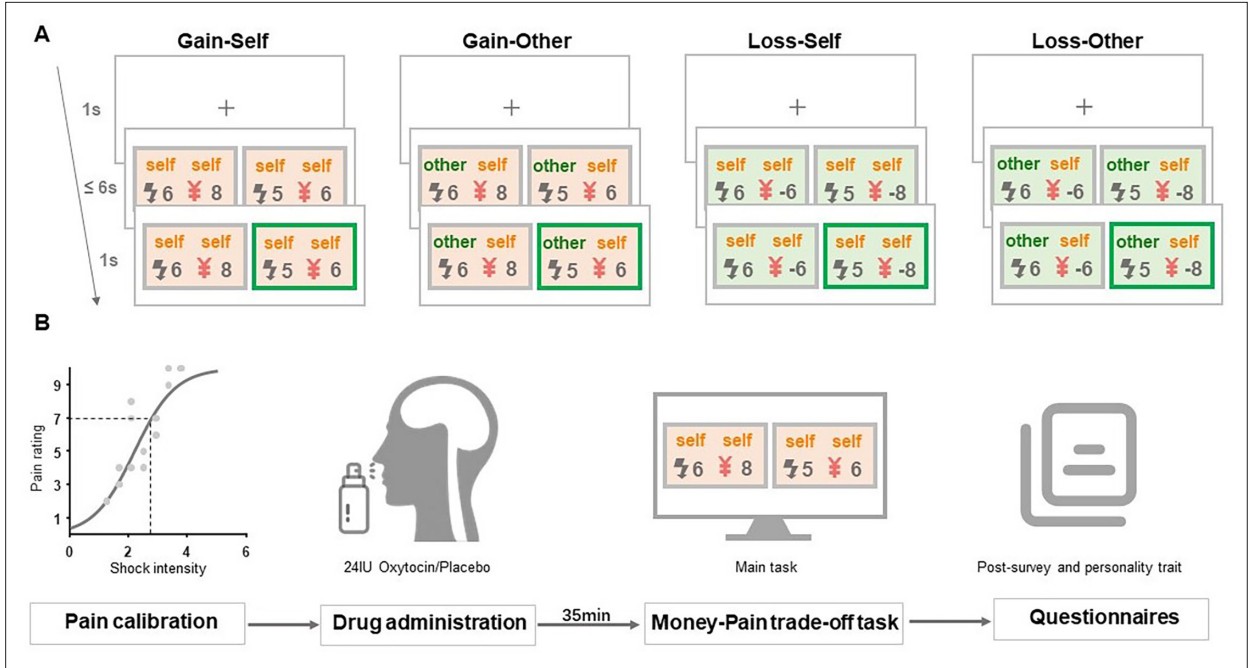

**Figure 1.** Experimental design and task. (**A**) Experimental task. Subjects performed a money-pain trade-off task in which they were designated as deciders. Four conditions (gain-self, gain-other, loss-self, loss-other) were introduced across decision contexts (gain vs. loss) and shock-recipients (self vs. other). Prior to the task, subjects received a neutral description of monetary values and potential shocks associated with choices options (see *Figure 1—figure supplement 1* for details). In each trial, subjects were asked to choose between two options with various amounts of monetary and harm consequences. The chosen option was highlighted for 1 s after subjects' decisions. (**B**) Procedures of the oxytocin study (study 2). Before the task, a pain calibration procedure was performed on each subject to determine their pain thresholds for electrical shock stimuli. Subjects were then administered with 24 IU oxytocin nasal spray or placebo (saline). Then, 35 min later, subjects commenced the money-pain trade-off task. Finally, they filled out questionnaires including post-task surveys and assessments of personality traits.

The online version of this article includes the following figure supplement(s) for figure 1:

**Figure supplement 1.** Experimental instructions provided to subjects.

cooperation and defensive aggression toward competing out-group members (*Zhang et al., 2019*; *De Dreu et al., 2010*; *De Dreu et al., 2020*). It is therefore plausible that oxytocin might also exert a context-dependent effect on hyperaltruistic preference in a moral decision task. Specifically, oxytocin might influence the way subjects internally frame the task as benefiting from others' pain or sacrificing self-interest to avoid others' harm given the widely reported link between oxytocin and empathy (*Barchi-Ferreira and Osório, 2021*; *Patin et al., 2018*; *Barraza and Zak, 2009*; *Stevens and Taber, 2021*). Also, previous literature on the prosocial effect of oxytocin dovetails nicely with the utilitarian moral literature suggesting that moral behavior is closely related to people's personality traits such as the instrumental harm attitude (IH), the degree to which people are willing to compromise their moral beliefs by inflicting harm on others to achieve better outcomes (*Kahane et al., 2018*).

We conducted two studies to test the above hypotheses. In study 1, we manipulated the decision context of the monetary valence (from making monetary gains to avoiding monetary losses) in a well-established money-pain trade-off task to examine the contextual specificity of the hyperaltruistic preferences (*Figure 1A*). We compared subjects' hyperaltruistic preferences in decision scenarios where options were constructed such that higher monetary gains (or smaller monetary loss in the loss context) were associated with more painful electric shocks (more painful option), and lower monetary gains (or bigger monetary loss in the loss context) were associated with less painful shocks (less painful option). We found that in line with results reported in previous studies, subjects showed clear hyperaltruistic preference in the gain context. However, shifting the decision context from gains to losses eliminated such preference. In study 2, we employed a pre-registered, placebo-controlled experimental design to examine how oxytocin might modulate the context effect of hyperaltruism (*Figure 1B*). We found that oxytocin had no effect on hyperaltruistic preference in the gain context. However, oxytocin

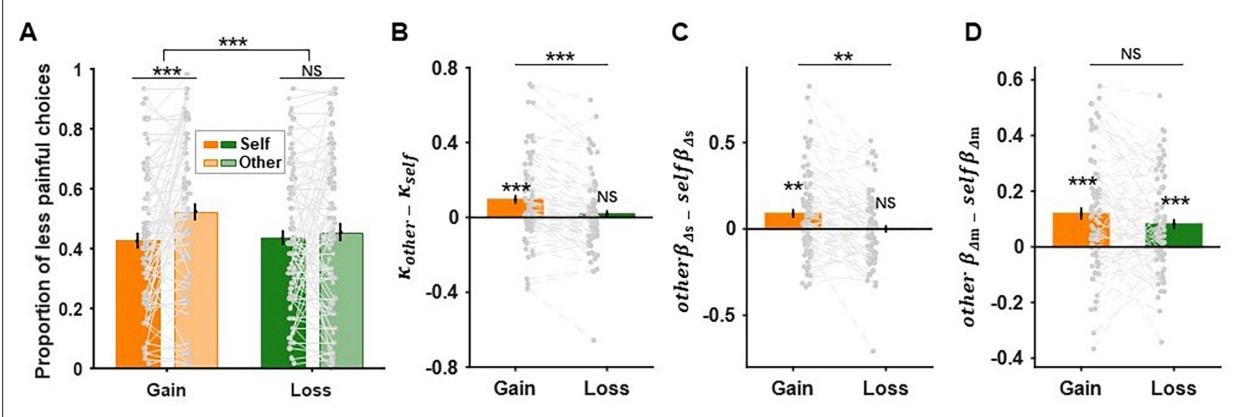

**Figure 2.** Context-specific hyperaltruistic preferences. (**A**) Subjects chose less painful options more frequently for others than for themselves in the gain context, demonstrating a hyperaltruistic preference. However, this tendency was absent in the loss context. (**B**) The harm aversion parameter κ for others was significantly greater than that of self in the gain context but not in the loss context (also see the 'Materials and methods' section and **Figure 2—figure supplement 2** for the harm aversion model analysis and model comparison results). (**C**) Furthermore, a mixed-effect logistic regression analysis showed that the relative harm sensitivity, calculated as the difference of regression coefficients of $\Delta s$ in the other- and self-conditions ($other\beta_{\Delta s} - self\beta_{\Delta s}$), was significant in the gain context, but not in the loss context (also see **Figure 2—figure supplement 1** for details). (**D**) However, the relative money sensitivity, the difference of regression coefficients of $\Delta m$ in the other- and self-conditions ($other\beta_{\Delta m} - self\beta_{\Delta m}$), did not show contextual specificity. Error bars represent SE across subjects (n=80). NS, not significant; ** P< 0.01 and *** P< 0.001.

The online version of this article includes the following figure supplement(s) for figure 2:

**Figure supplement 1.** Mixed-effect logistic regression analysis results for study 1 (regression model 1).

**Figure supplement 2.** Model comparison results.

restored subjects' hyperaltruistic preferences in the loss context. Importantly, oxytocin administration prompted subjects to more likely perceive the task structure as harming others, which in turn mediated their hyperaltruistic preferences.

## Results

Prior to the money-pain trade-off task, we individually calibrated each subject's pain threshold using a standard procedure (*Crockett et al., 2014*; *Crockett et al., 2015*; *Crockett et al., 2017*). This allowed us to tailor a moderate electric stimulus that corresponded to each subject's subjective pain intensity. Subjects then engaged in 240 decision trials (60 trials per condition), acting as the 'decider' and trading off between monetary gains or losses for themselves and the pain experienced by either themselves or an anonymous 'pain receiver' (gain-self, gain-other, loss-self, and loss-other, see *Figure 1—figure supplement 1* and 'Materials and methods' section for details).

### Loss context eliminated hyperaltruistic preference

We first tested whether the decision context (gain or loss) directly affected subjects' hyperaltruistic tendencies by examining the proportion of trials in which subjects chose the less painful option both for themselves and for others (*Figure 1A*). Since the choice sets for the self- and other-recipient are the same, subjects' hyperaltruistic tendencies can be examined by the choice differences regarding different shock recipients (self vs. other). A larger choice difference between other- and self-recipients indicates a stronger hyperaltruistic tendencies in both the gain and loss contexts. The main effects of the shock recipient (self vs. other) and decision context (gain vs. loss) were both significant (recipient: $F_{1,79}= 6.625, P = 0.012, \eta^2 = 0.077$; context: $F_{1,79} = 8.379, P = 0.005, \eta^2 = 0.096$), indicating that subjects were in general hyperaltruistic across decision contexts and more likely to choose the more painful option in the loss context (*Figure 2A*). There was also a significant recipient × context interaction effect $F_{1,79} = 33.047, P < 0.001, \eta^2 = 0.295$, suggesting that the loss context (vs. gain context) significantly decreased subjects' hyperaltruistic tendencies (*Figure 2A*). Further simple effect analysis confirmed that hyperaltruism only existed in the gain context ($F_{1,79}= 16.798, P< 0.001, \eta^2 = 0.175$) but was eliminated in the loss context ($F_{1,79}= 0.650, P= 0.423, \eta^2 = 0.008$). We also modeled subjects'

choices using an influential model where subjects' behavior could be characterized by the harm (electric shock) aversion parameter κ, reflecting the relative weights subjects assigned to $\Delta m$ and $\Delta s$, the objective difference in money and shocks between the more and less painful options, respectively ($\Delta V = (1 - \kappa)\Delta m - \kappa\Delta s$; *Equation 1*, see 'Materials and methods' for details) (*Crockett et al., 2014*; *Crockett et al., 2015*; *Crockett et al., 2017*). Higher κ indicates that higher weight is assigned to $\Delta s$ than $\Delta m$ and vice versa. Consistent with previous literature, we found that the harm aversion parameter κ for the other-recipient was significantly greater than that of the self-recipient in the gain context (*Crockett et al., 2014*; *Crockett et al., 2015*; *Crockett et al., 2017*); however, such harm aversion asymmetry between self- and other-recipient disappeared in the loss context ($\kappa_{other} - \kappa_{self} > 0$ in the gain context: $t_{79} = 3.869, P < 0.001$; $\kappa_{other}$ and $\kappa_{self}$ showed no difference in the loss context: $t_{79} = 0.834, P = 0.407$; $\kappa_{other} - \kappa_{self}$ showed context difference: $t_{79} = 5.591, P < 0.001$; *Figure 2B*).

To further identify the distinct effects of $\Delta m$ and $\Delta s$ in biasing subjects' choice behavior, we also ran a mixed-effect logistic regression analysis to examine how the change of decision context (gain vs. loss) affected each individual's sensitivities to money ($\Delta m$) and electric shock ($\Delta s$). Subjects' choices were regressed against $\Delta m$ and $\Delta s$, and this analysis yielded significant effects on relative harm sensitivity ($other\,\beta_{\Delta s} - self\,\beta_{\Delta s}$: difference of regression coefficients of $\Delta s$ in the other- and self-conditions) in the gain context but not in the loss context (gain context: $t_{79} = 3.298, P = 0.001$; loss context: $t_{79} = 0.015, P = 0.988$) and significant decision context effect ($t_{79} = 3.630, P = 0.001$; *Figure 2C*, also see *Figure 2—figure supplement 1* for detailed results of this regression analysis). On the contrary, decision context did not have an effect on subjects' relative sensitivities towards monetary ($\Delta m$) difference ($t_{79} = 1.480, P = 0.143$), though the relative money sensitivities ($other\beta_{\Delta m} - self\beta_{\Delta m}$: difference of regression coefficients of $\Delta m$ in the other- and self-condition) were significant in both contexts (gain context: $t_{79} = 5.241, P < 0.001$; loss context: $t_{79} = 4.355, P < 0.001$; *Figure 2D*). These results suggest that, contrary to the prediction of loss aversion, the relative money sensitivity did not change when the task structure switched from monetary gains to losses. Instead, subjects' susceptibilities to harm might underlie the diminishment of hyperaltruism across decision contexts.

## Individual differences in moral preference

Recent theoretical development in moral decision-making stresses the importance of distinctive dimensions of IH and impartial beneficence (IB) in driving people's moral preference (*Awad et al., 2020*; *Everett et al., 2021*). The IH and IB components of the Oxford utilitarianism scale (*Kahane et al., 2018*) of moral psychology were used to measure the extent to which individuals are willing to compromise their moral beliefs by breaking the rules or inflicting harm on others to achieve better outcomes (IH) and the extent of the impartial concern for the well-being of everyone (IB), respectively. We tested how hyperaltruism was related to both IH and IB across decision contexts using an exploratory multiple regression analysis. Moral preference, defined as $\kappa_{others} - \kappa_{self}$, was negatively associated with IH ($\beta = -0.031 \pm 0.011$, $t_{156} = -2.784, P = 0.006$) but not with IB ($\beta = 0.008 \pm 0.016$, $t_{156} = 0.475, P = 0.636$) across gain and loss contexts, reflecting a general connection between moral preference and IH (*Figure 3A and B*). Note that the interaction effects of context × IH and context × IB were not significant (context × IH effect: $\beta = 0.016 \pm 0.022$, $t_{152} = 0.726, P = 0.469$, *Figure 3A*; context × IB effect: $\beta = -0.004 \pm 0.031$, $t_{152} = -0.115, P = 0.908$, *Figure 3B*). Interestingly, by examining the separate contributions of $\Delta m$ and $\Delta s$ to moral preferences, we found that the decision context modulated the relationship between IH and subjects' relative harm sensitivities (gain context: $\beta = -0.053 \pm 0.016, t_{152} = -3.224, P = 0.002$; loss context: $\beta = -0.007 \pm 0.016, t_{152} = -0.453, P = 0.651$; context × IH: $\beta = 0.045 \pm 0.023, t_{152} = 1.959, P = 0.651$; *Figure 3C*), yet it did not modulate the relationship between IB and subjects' relative harm sensitivities (gain context: $\beta = 0.019 \pm 0.023, t_{152} = 0.804$, $P = 0.423$; loss context: $\beta = 0.010 \pm 0.023, t_{152} = 0.425, P = 0.672$; context × IB: $\beta = -0.009 \pm 0.033$, $t_{152} = -0.268, P = 0.789$; *Figure 3—figure supplement 2A*). However, subjects' monetary sensitivities were not related to IH (gain context: $\beta = -0.022 \pm 0.015, t_{152} = -1.476, P = 0.142$; loss context: $\beta = -0.023 \pm 0.015, t_{152} = -1.523, P = 0.130$; context × IH: $\beta = -0.001 \pm 0.021, t_{152} = -0.034, P = 0.973$; *Figure 3D*), nor with IB (gain context: $\beta = -0.008 \pm 0.021, t_{152} = -0.369, P = 0.713$; loss context: $\beta = -0.009 \pm 0.021, t_{152} = -0.414, P = 0.679$; context × IB: $\beta = -0.001 \pm 0.030, t_{152} = -0.032, P = 0.974$; *Figure 3—figure supplement 2B*, also see *Figure 3—figure supplement 1* and *Table 1*). These results suggest that the context-dependent moral preference may be specifically due to the context modulation of subjects' relative harm sensitivities, accompanied by the altered correlation

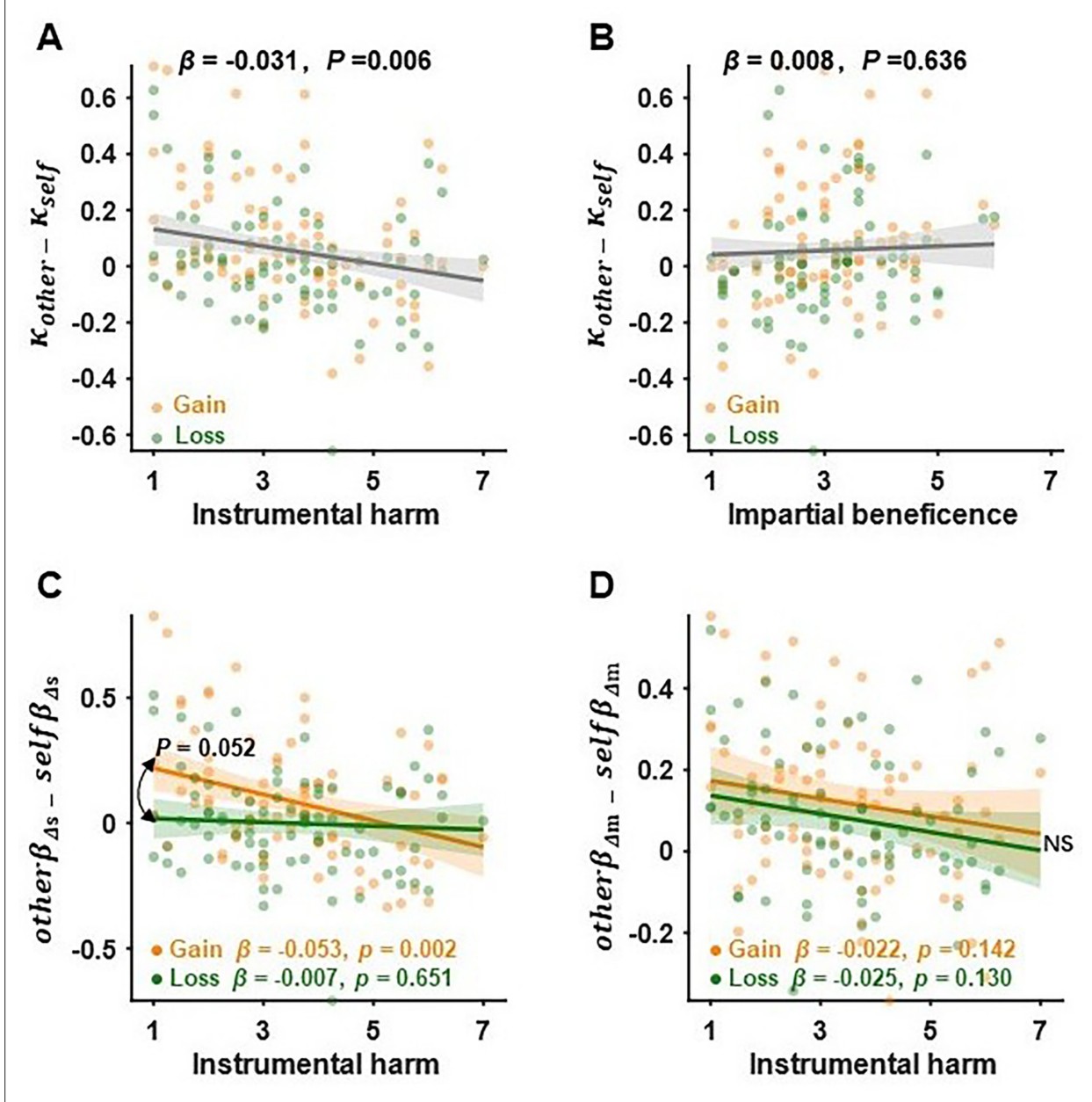

**Figure 3.** Individual difference in moral preferences. Hyperaltruism ($\kappa_{other} - \kappa_{self}$) was negatively associated with instrumental harm (IH) attitudes (**A**) but not with impartial beneficence (IB) attitudes (**B**). (**C**) The correlation between IH and subjects' relative harm sensitivities ($other\beta_{\Delta s} - self\beta_{\Delta s}$) was marginally different between the gain and loss contexts (also see *Figure 3—figure supplement 2A*). (**D**) However, no significant associations were observed between subjects' relative monetary sensitivities ($other\beta_{\Delta m} - self\beta_{\Delta m}$) and IH or IB (see *Figure 3—figure supplement 2B*). Please note that since both IH and IB were correlated with empathic concern (EC) scores (see *Figure 3—figure supplement 1*), the multiple regression analysis was conducted with EC included as a covariate of no interest. The regression coefficients in (**A, C, D**) show the relationship between IH and moral behavior, controlling for EC and IB, while the coefficient in (**B**) shows the relationship between IB and moral preference, controlling for EC and IH. NS, not significant.

The online version of this article includes the following figure supplement(s) for figure 3:

**Figure supplement 1.** The relationship between empathic concern (EC) and utilitarian moral personality traits.

**Figure supplement 2.** Association between impartial beneficence (IB) and subjects' relative harm/money sensitivities in Study 1.

**Table 1.** Contribution of empathic concern (EC), instrumental harm (IH), and impartial beneficence (IB) scores on subjects' behavior in study 1.

The table presents results from three multiple regression analyses. The dependent variables in the three analyses were hyperaltruism ($\kappa_{other} - \kappa_{self}$), relative harm sensitivity ($other\beta_{\Delta s} - self\beta_{\Delta s}$), and relative monetary sensitivity ($other\beta_{\Delta m} - self\beta_{\Delta m}$). Results in bold are further illustrated in *Figure 3C*.

| Predictors | Hyperaltruism | | | | Relative harm sensitivity | | | | Relative monetary sensitivity | | | |
|---|---|---|---|---|---|---|---|---|---|---|---|---|
| | β | SE | t | P | β | SE | t | P | β | SE | t | P |
| Intercept | –0.086 | 0.153 | –0.563 | 0.574 | –0.086 | 0.162 | –0.532 | 0.596 | 0.121 | 0.147 | 0.825 | 0.411 |
| EC | 0.011 | 0.006 | 2.080 | 0.039 | 0.012 | 0.006 | 2.063 | 0.041 | 0.004 | 0.005 | 0.738 | 0.461 |
| IH | –0.039 | 0.015 | –2.501 | 0.013 | –0.053 | 0.016 | –3.224 | 0.002 | –0.022 | 0.015 | –1.476 | 0.142 |
| IB | 0.009 | 0.022 | 0.421 | 0.675 | 0.019 | 0.023 | 0.804 | 0.423 | –0.008 | 0.021 | –0.369 | 0.713 |
| Context (gain vs. loss) | –0.163 | 0.217 | –0.754 | 0.452 | –0.298 | 0.229 | –1.302 | 0.195 | 0.058 | 0.208 | 0.280 | 0.780 |
| EC × context | 0.002 | 0.008 | 0.216 | 0.829 | 0.003 | 0.008 | 0.383 | 0.702 | –0.004 | 0.007 | –0.484 | 0.629 |
| IH × context | 0.016 | 0.022 | 0.726 | 0.469 | **0.045** | **0.023** | **1.959** | **0.052** | –0.001 | 0.021 | –0.034 | 0.973 |
| IB × context | –0.004 | 0.031 | –0.115 | 0.908 | –0.009 | 0.033 | –0.268 | 0.789 | –0.001 | 0.030 | –0.032 | 0.974 |

between relative harm sensitivities and IH across decision contexts. Furthermore, our results are consistent with the claim that profiting from inflicting pains on another person (IH) is inherently deemed immoral (*Kahane et al., 2018*). Hyperaltruistic preference, therefore, is likely to be associated with subjects' IH dispositions.

## Oxytocin restored hyperaltruistic preferences in the loss context

To probe how oxytocin might alter subjects' moral preference, we conducted the preregistered study 2 where a separate cohort of subjects performed the same task while undergoing the placebo/oxytocin administration in a with-subject design (N=46, see 'Materials and methods' for experimental details). First, in the placebo session of study 2, we replicated the major findings of study 1 (*Figure 2A*), demonstrating that hyperaltruistic preference was not significant in the loss context compared to the gain context (Placebo: context × recipient interaction: $F_{1,45} = 9.103, P = 0.004, \eta^2 = 0.168$; simple effect: gain context $F_{1,45} = 9.356, P = 0.004, \eta^2 = 0.172$; loss context $F_{1,45} = 0.005, P = 0.946, \eta^2 < 0.001$; *Figure 4A*). The main effect of treatment (placebo vs. oxytocin) was significant ($F_{1,45} = 41.961, P < 0.001, \eta^2 = 0.483$), indicating that the oxytocin treatment prompted subjects to select the more painful option more often. Interestingly, the treatment × recipient × context interaction effect was also significant ($F_{1,45} = 6.309, P = 0.016, \eta^2 = 0.483$), suggesting that the recipient × context interaction might be different due to the placebo and oxytocin treatments. Indeed, with the administration of oxytocin, the hyperaltruistic preference was restored in the loss context, without affecting the hyperaltruistic pattern in the gain context (oxytocin: context × recipient: $F_{1,45} = 0.480, P = 0.492, \eta^2 = 0.011$ simple effect: gain context: $F_{1,45} = 25.364, P < 0.001, \eta^2 = 0.360$ ; loss context: $F_{1,45} = 24.408, P < 0.001, \eta^2 = 0.352$; *Figure 4A*). These results together highlight that oxytocin might play an important context-dependent modulatory role in restoring hyperaltruistic moral preference. Our modeling analysis revealed similar results: there was a significant treatment × context interaction effect ($F_{1,45} = 6.349, P = 0.015, \eta^2 = 0.124$) on subjects' moral preference (defined as $\kappa_{other}$-$\kappa_{self}$, *Figure 4B*). With the administration of oxytocin, however, the diminished hyperaltruistic preference in the loss context (placebo: $F_{1,45} = 1.295, P = 0.261, \eta^2 = 0.028$) was successfully restored (oxytocin: $F_{1,45} = 17.990, P < 0.001, \eta^2 = 0.286$, simple effect analysis, *Figure 4B*). Additionally, the model-based moral preference was correlated with subjects' IH reports (but not IB) across subjects in the placebo treatments (*Figure 4—figure supplements 1 and 2*, also see *Table 2*), replicating the results we obtained in study 1 (*Figure 3A and B*).

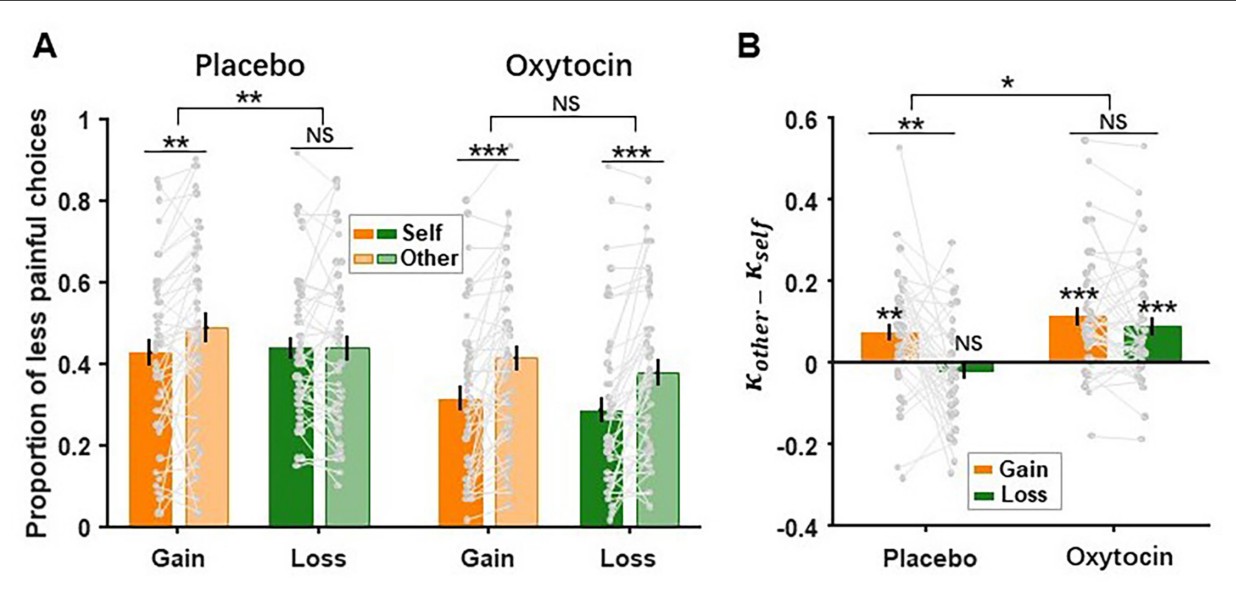

**Figure 4.** Oxytocin significantly promoted hyperaltruistic preference in the loss context. (**A**) In contrast to the placebo condition, oxytocin administration restored hyperaltruistic behavior in the loss decision context. (**B**) Model-based hyperaltruistic parameter ($\kappa_{other} - \kappa_{self}$) showed similar patterns: hyperaltruistic tendency was reduced in the loss context of the placebo session but restored in the oxytocin session (also see *Figure 4—figure supplements 1 and 2* for the influence of oxytocin on the persoanlity triats and the relationship between hyperaltruistic preference and these moral personality triats). Error bars represent SE across subjects (n=46). NS, not significant; * $P < 0.05$, ** $P < 0.01$, and *** $P < 0.001$.

The online version of this article includes the following figure supplement(s) for figure 4:

**Figure supplement 1.** Oxytocin did not influence personality traits in study 2.

**Figure supplement 2.** Relationships between hyperaltruism and utilitarian moral personality traits in study 2.

## Hyperaltruistic preference and increased harm sensitivities with oxytocin administration

In study 1, we showed that the lack of hyperaltruism in the loss context was specifically related to the diminished relative harm sensitivities. In study 2, we also ran a mixed-effect logistic regression of subjects' choices against continuous independent variables including Δm, Δs and categorical variables including treatment (placebo vs. oxytocin), harm/shock recipient (self vs. other), and decision context valence (gain vs. loss; see 'Materials and methods' for details). This regression analysis yielded a significant Δs × treatment × recipient × context interaction effect ($\beta = 0.125 \pm 0.053, P = 0.018, 95\%\text{CI} = [0.022, 0.228]$; *Figure 5—figure supplements 1 and 2*), indicating the relative harm sensitivities ($other\beta_{\Delta s} - self\beta_{\Delta s}$) were modulated by the specific treatment and decision combinations. Indeed, repeated measures ANOVA confirmed that in the placebo session, as in study 1, the relative harm sensitivities decreased in the loss context (relative to the gain context, simple effect: $F_{1,45} = 11.521, P = 0.001, \eta^2 = 0.204$; *Figure 5A*); however, with the administration of oxytocin, there was no significant difference of the relative harm sensitivities between the gain and loss contexts (simple effect: $F_{1,45} = 0.131, P = 0.719, \eta^2 = 0.001$; *Figure 5A*). It is worth noting that oxytocin did not alter the gain/loss relative money sensitivity difference (context × treatment interaction: $F_{1,45} = 1.933, P = 0.171, \eta^2 = 0.041$; *Figure 5B*), which corresponded to a non-significant Δm × treatment × recipient × context interaction effect ($\beta = -0.052 \pm 0.100, P = 0.601, 95\%\text{CI} = [-0.249, 0.144]$; *Figure 5—figure supplement 1*). Furthermore, in the placebo session, the decision context significantly modulated the relationship between the relative harm sensitivity and IH (gain context: $\beta = -0.071 \pm 0.024, t_{84} = -3.005, P = 0.004$; loss context: $\beta = 0.006 \pm 0.024, t_{84} = -0.243, P = 0.809$; context × IH: $\beta = 0.077 \pm 0.033, t_{84} = 2.297, P = 0.024$; *Figure 5C*). However, under the treatment of oxytocin, the context × IH interaction became nonsignificant ($\beta = -0.006 \pm 0.021, t_{84} = -0.272, P = 0.786$; *Figure 5D*), despite the significant negative relationship between the relative harm sensitivities and IH in both decision contexts (gain context: $\beta = -0.037 \pm 0.015, t_{84} = -2.458, P = 0.016$; loss context: $\beta = -0.042 \pm 0.015, t_{84} = -2.843, P = 0.006$; *Figure 5D*) (also see *Figure 5—figure supplement 3* and *Table 2*). The effect size (Cohen's $f^2$) for

**Table 2.** Contribution of empathic concern (EC), instrumental harm (IH), and impartial beneficence (IB) scores on participants' behavior in study 2.

Tables 2a (placebo session) and 2b (oxytocin session) show the results of three multiple regression analyses. The dependent variables in the three analyses were hyperaltruism ($\kappa_{other} - \kappa_{self}$), relative harm sensitivity ($other\beta_{\Delta s} - self\beta_{\Delta s}$), and relative monetary sensitivity ($other\beta_{\Delta m} - self\beta_{\Delta m}$). Results in bold are further illustrated in *Figure 5C and D*.

a. Placebo condition

| Predictors | Hyperaltruism | | | | Relative harm sensitivity | | | | Relative monetary sensitivity | | | |
|---|---|---|---|---|---|---|---|---|---|---|---|---|
| | β | SE | t | P | β | SE | t | P | β | SE | t | P |
| Intercept | −0.085 | 0.123 | −0.688 | 0.493 | 0.065 | 0.17 | 0.382 | 0.704 | −0.205 | 0.266 | −0.771 | 0.443 |
| EC | 0.005 | 0.004 | 1.162 | 0.248 | 0.006 | 0.006 | 1.086 | 0.281 | −0.004 | 0.009 | −0.444 | 0.658 |
| IH | −0.032 | 0.017 | −1.904 | 0.06 | −0.071 | 0.024 | −3.005 | 0.003 | 0.035 | 0.037 | 0.967 | 0.336 |
| IB | 0.047 | 0.021 | 2.2 | 0.031 | 0.029 | 0.029 | 1 | 0.32 | 0.086 | 0.046 | 1.887 | 0.063 |
| Context (gain vs. loss) | −0.034 | 0.174 | −0.197 | 0.844 | −0.34 | 0.241 | −1.412 | 0.162 | 0.323 | 0.375 | 0.862 | 0.391 |
| EC × context | 0.002 | 0.006 | 0.332 | 0.74 | 0.005 | 0.008 | 0.601 | 0.549 | 0.003 | 0.012 | 0.264 | 0.792 |
| IH × context | 0.018 | 0.024 | 0.737 | 0.463 | **0.077** | **0.033** | **2.297** | **0.024** | −0.062 | 0.052 | −1.187 | 0.238 |
| IB × context | −0.051 | 0.03 | −1.711 | 0.091 | −0.047 | 0.042 | −1.14 | 0.257 | −0.084 | 0.065 | −1.295 | 0.199 |
| b. Oxytocin condition | | | | | | | | | | | | |
| Intercept | 0.138 | 0.125 | 1.1 | 0.275 | 0.056 | 0.109 | 0.511 | 0.611 | 0.049 | 0.253 | 0.195 | 0.846 |
| EC | 0.01 | 0.004 | 2.397 | 0.019 | 0.006 | 0.004 | 1.675 | 0.098 | 0.001 | 0.009 | 0.157 | 0.876 |
| IH | −0.047 | 0.017 | −2.721 | 0.008 | −0.037 | 0.015 | −2.458 | 0.016 | 0.053 | 0.035 | 1.541 | 0.126 |
| IB | −0.034 | 0.019 | −1.813 | 0.073 | −0.009 | 0.016 | −0.582 | 0.562 | −0.041 | 0.038 | −1.091 | 0.279 |
| Context (gain vs. loss) | −0.071 | 0.177 | −0.401 | 0.69 | 0.092 | 0.154 | 0.596 | 0.553 | −0.372 | 0.358 | −1.04 | 0.301 |
| EC × context | 0.002 | 0.006 | 0.385 | 0.701 | −0.002 | 0.005 | −0.336 | 0.737 | 0.012 | 0.012 | 0.985 | 0.327 |
| IH × context | 0.008 | 0.024 | 0.323 | 0.747 | **−0.006** | **0.021** | **−0.272** | **0.786** | −0.045 | 0.049 | −0.918 | 0.361 |
| IB × context | −0.01 | 0.026 | −0.384 | 0.702 | −0.012 | 0.023 | −0.526 | 0.6 | 0.024 | 0.053 | 0.457 | 0.649 |

this exploratory analysis was calculated to be 0.491 and 0.379 for the placebo and oxytocin conditions, respectively. The post hoc power analysis with a significance level of $\alpha=0.05$, 7 regressors (IH, IB, EC, decision context, IH × context, IB × context, and EC ×context), and sample size of N=46 yielded achieved power of 0.910 (placebo treatment) and 0.808 (oxytocin treatment). These findings further highlighted the influence of decision context on hyperaltruistic moral preference and stressed the importance of oxytocin in restoring the correlation between the relative harm sensitivity and the dispositional personality traits such as IH.

### Oxytocin eliminated the modulation effect of decision context on hyperaltruism

We reasoned that the decision context effect of hyperaltruistic preferences might be rooted in how our subjects internally framed the task as 'harming' (inflicting pain on the receiver to increase monetary gain/avoid monetary loss) or 'helping' others (sacrificing greater monetary gain/accepting greater monetary loss to alleviate the receiver's pain). To test this hypothesis, in study 2 we asked subjects to subjectively report whether they perceived the task as 'harmful' or 'helpful' (see 'Materials and methods' section for more details) in both placebo and oxytocin sessions (gain and loss contexts for each session). The non-parametric Friedman tests on subjects' harm framing report showed a significant difference in gain and loss contexts under placebo condition ($X^2 = 2.827, P = 0.028$, Bonferroni correction; *Figure 6A*), suggesting that subjects were more likely to perceive the task as harming others in the gain context. In addition, the administration of oxytocin increased subjects' perception of causing harm to others in the loss context ($X^2 = 2.665, P = 0.046$, Bonferroni correction) and removed contextual disparities in harm perception (context × treatment interaction: $X^2 = 7.410, P = 0.006$)

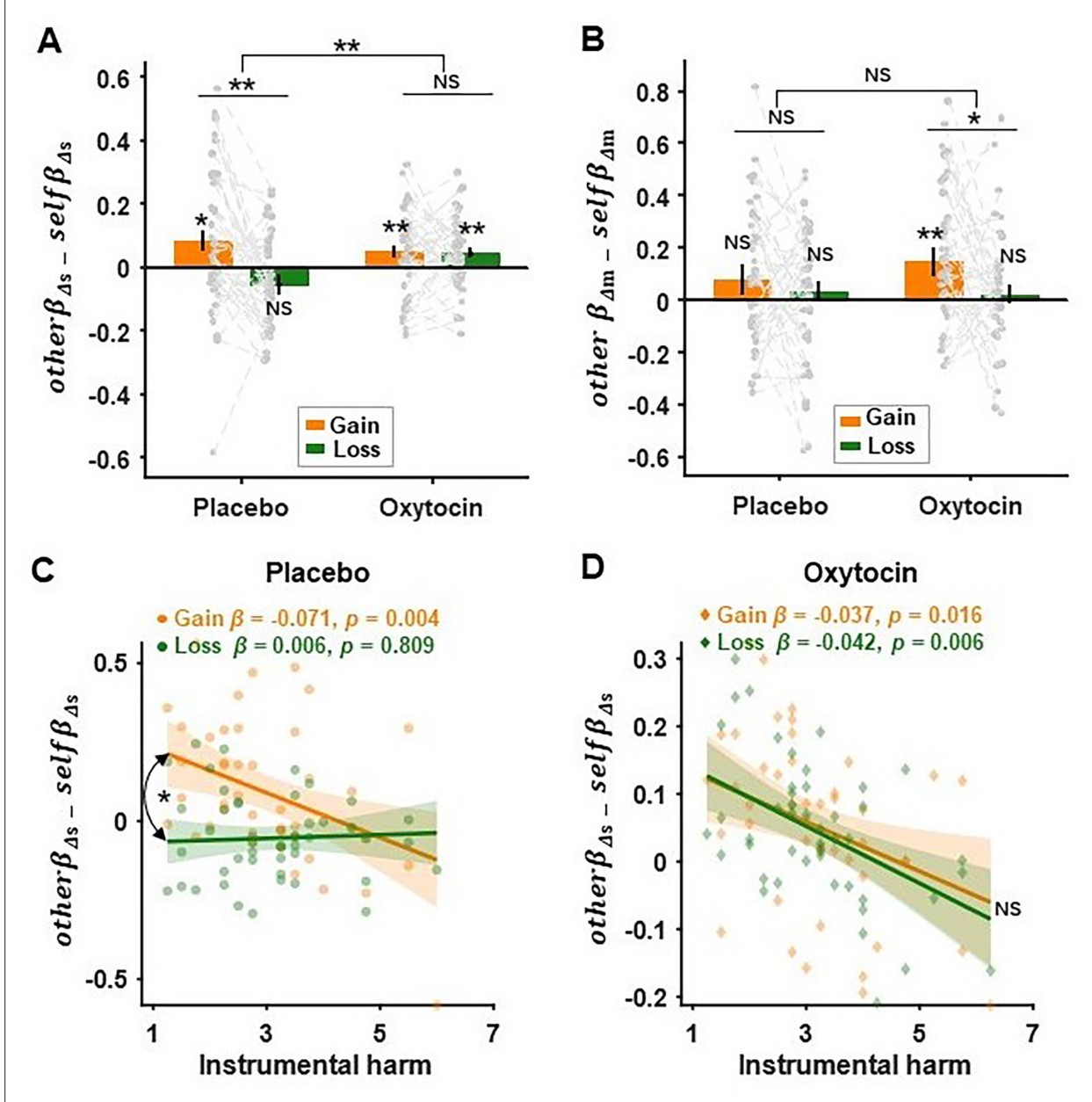

**Figure 5.** Oxytocin effect on relative harm/ money sensitivities. (**A, B**) The effect of oxytocin on relative harm/ money sensitivity was assessed via the mixed-effect logistic regression analysis (regression model 2). Oxytocin significantly modulated the context-specificity of relative harm sensitivity ($other\beta_{\Delta s} - self\beta_{\Delta s}$), while having no effect on the contextual differences of the relative monetary sensitivities ($other\beta_{\Delta m} - self\beta_{\Delta m}$) (also see *Figure 5—figure supplements 1 and 2* for detailed results of the regression analysis). (**C**) The decision context modulated the correlation between instrumental harm (IH) and subjects' harm sensitivities in the placebo session. (**D**) The modulation effect of decision context was absent in the oxytocin session (also see *Figure 5—figure supplement 3* for the oxytocin's effect on the association between relative monetary sensitivities and IH). Error bars represent SE across subjects. The regression coefficients in (**C, D**) showed the association between IH and relative harm sensitivities, controlling for empathic concern and impartial beneficence. NS, not significant; * $P < 0.05$, ** $P < 0.01$.

The online version of this article includes the following figure supplement(s) for figure 5:

**Figure supplement 1.** The main results of mixed-effect logistic regression analysis in study 2 (regression model 2).

**Figure supplement 2.** Pain sensitivity across experimental conditions.

**Figure supplement 3.** Associations between relative harm/money sensitivities and utilitarian moral personality traits in study 2.

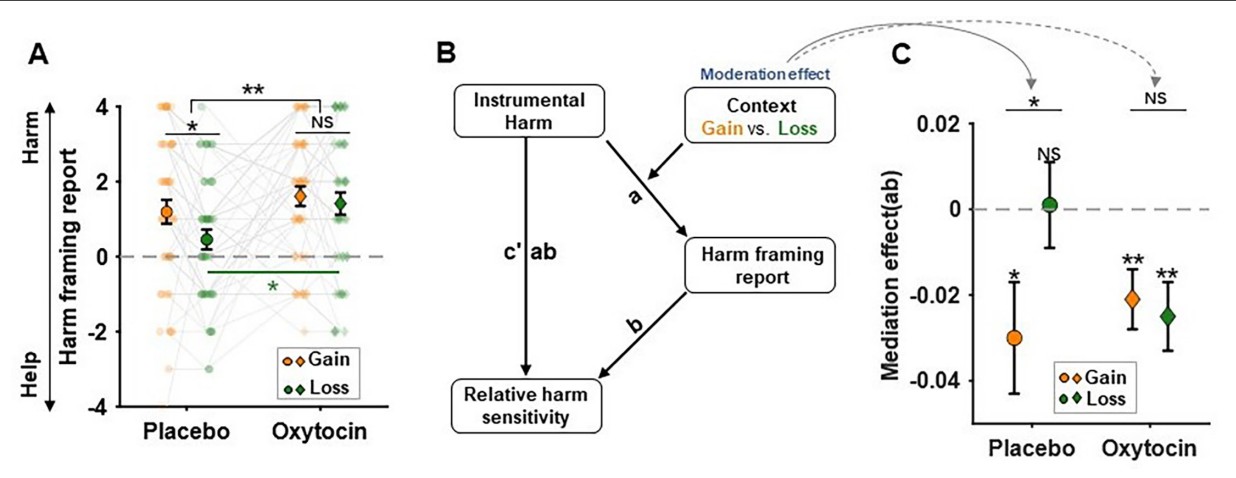

**Figure 6.** Oxytocin modulated the contextual influence on hyperaltruistic behaviors. (**A**) The non-parametric Friedman tests showed that monetary loss (relative to gain) significantly reduced subjects' perception of harm framing in the task. Oxytocin augmented harm framing perception, particularly in the loss context, effectively removing the contextual specificity of harm framing perception. (**B**) The conceptual diagram of the moderated mediation model. We assume that the perceived harm framing mediates the relationship between instrumental harm and relative harm sensitivity, with decision context moderating the mediation effect. (**C**) The moderating effect of decision context was significant under placebo condition. However, oxytocin obliterated the contextual moderation effect by reinstating the mediating role of harm perception in the loss context. Error bars represent SE across subjects. NS, not significant; * $P < 0.05$, ** $P < 0.01$.

(*Figure 6A*), suggesting that the oxytocin administration successfully erased the framing difference between the gain and loss decision contexts.

We further hypothesized that the correlations between IH personality traits and subjects' differential harm sensitivities ($other\beta_{\Delta s} - self\beta_{\Delta s}$, *Figure 5C and D*) might be mediated by how strongly subjects perceived the decision task as harming or helping others. A moderated mediation analysis was thus conducted to directly test this hypothesis. In the moderated mediation model, the effects of decision context can be expressed as the moderation effects on mediation (*Figure 6B*). As we expected, in the placebo session, the moderating effect of decision context was significant (placebo: the differential mediation effect in gain and loss contexts: $\Delta ab = -0.036, P = 0.036, 95\%CI = [-0.073, -0.003]$; *Figure 6C*). Specifically, the mediation effect of

**Table 3.** Moderated mediation analysis results for study 2.

In the moderated mediation model, IH serves as the independent variable, harm framing report as the mediator, relative harm sensitivity as the dependent variable, and decision context (gain vs. loss) as the moderator. Coefficient **a** represents the influence of the independent variable on the mediator, while coefficient **b** represents the effect of the mediator on the dependent variable, and coefficient **c'** represents the direct effect (insignificant **c'** suggests a full mediation effect). The coefficients in bold indicate the moderation effect of the context (gain vs. loss) in both the placebo and oxytocin sessions *$P<0.05$, **$P<0.01$, and ***$P<0.001$.

| | Placebo | | Oxytocin | |
|---|---|---|---|---|
| | Mediator | Dependent variable | Mediator | Dependent variable |
| Predictors | Harm framing report β (SE) | Relative harm sensitivity β (SE) | Harm framing report β (SE) | Relative harm sensitivity β (SE) |
| Intercept | 1.196 (0.279) *** | 0.069 (0.023) ** | 1.609 (0.239) *** | 0.049 (0.014) *** |
| Context (gain vs. loss) | –0.739 (0.396) | –0108 (0.032) ** | –0.196 (0.338) | –0.002 (0.019) |
| Empathic concern (EC) | 0.020 (0.046) | 0.008 (0.004) * | 0.023 (0.040) | 0.005 (0.002) * |
| Impartial beneficence (IB) | –0.116 (0.239) | 0.010 (0.019) | –0.270 (0.179) | –0.008 (0.010) |
| Instrumental harm (IH) | –0.758 (0.257) *** a | –0.018 (0.016) c' | –0.753 (0.220) *** a | –0.017 (0.011) c' |
| Context × IH | **0.784 (0.343) *** | – | **–0.137 (0.291)** | – |
| Harm framing report | – | 0.039 (0.008) *** b | – | 0.028 (0.006) *** b |

perceived harm was significant in the gain context (ab $= -0.030, P = 0.021, 95\%\text{CI} = [-0.056, -0.006]$) but not in the loss context (ab $= 0.001, P = 0.915, 95\%\text{CI} = [-0.019, 0.019]$). However, oxytocin eliminated the contextual moderation effect (oxytocin: the differential mediation effect in gain and loss contexts: $\Delta$ab $= 0.006\, P = 0.731, 95\%\text{CI} = [-0.044, 0.033]$; *Figure 6C*) by reinstating the mediation role of harm perception in the loss context (ab = -0.025, P= 0.002, 95% CI = [-0.042, -0.011]) (also see *Table 3*). These results indicated that subjects' perception of the task structure as helping or harming others mediated the correlation between personality trait IH and subjects' hyperaltruistic preference. Switching the decision context from gains to losses directly modulated subjects' moral perception of the task and nullified hyperaltruism. Finally, oxytocin administration restored participants' hyperaltruistic preference by reinstating the mediation role of perceived harm report on the correlation between IH and hyperaltruism across gain and loss decision contexts.

## Discussion

Utilitarian moral decision-making studies often involve moral dilemmas where utilitarian gains are traded against moral norms such as no deception or no infliction of suffering on others (*Kahane et al., 2018*; *Greene et al., 2001*; *Zhu et al., 2014*). Recent studies extended this line of research by comparing subjects' sensitivities toward others' suffering relative to their own suffering and revealed an intriguing hyperaltruistic preference: people were willing to forgo more monetary gain to reduce others' pain than their own pain (*Crockett et al., 2014*; *Crockett et al., 2015*; *Crockett et al., 2017*; *Volz et al., 2017*). In two experiments, we replicated findings in the previous literature (*Figures 2 and 4*). More importantly, we further showed that hyperaltruistic preference was susceptible to the corresponding decision context. Replacing the trade-off between monetary gain and electric shocks (gain context) with arbitration between monetary loss and electric shocks (loss context) eliminated subjects' hyperaltruistic preference that would otherwise have been observed. Importantly, oxytocin restored the hyperaltruistic preference in the loss context by biasing subjects to more likely perceive the decision context as harming others for self-interest. Finally, moral framing, or how subjects perceive the decision context as helping or harming others, mediated the association between the IH personality trait and subjects' sensitivities to others' suffering relative to their own suffering. Therefore, we demonstrated that both the hyperaltruistic preference and the effects of oxytocin were context-dependent and provided a mechanistic account and the boundary condition for hyperaltruistic disposition.

Recent development in utilitarian moral psychology highlights two independent dimensions that collectively shape people's moral preference: attitudes toward IH, the suffering of other individuals to achieve greater good, and toward IB, an impartial concern for the well-being of everyone (*Kahane et al., 2018*). In our experiments, we found that subjects' IH attitude was negatively associated with their hyperaltruistic preference (*Figure 3A*). However, there was no significant relationship between IB attitudes and hyperaltruism (*Figure 3B*), suggesting the money-pain trade-off task might be better suited to study the specific relationship between IH attitude and moral preference (*Greene et al., 2001*). Further analyses decomposing subjects' hyperaltruistic preference into their relative sensitivities toward shock difference ($\Delta s$) and money difference ($\Delta m$) suggested that the decision context (loss vs. gain) altered the relationship between IH and $\Delta s$ relative sensitivity (*Figures 3C and 5C*) but not the association between IH and $\Delta m$ relative sensitivity (*Figure 3D*). These results confirm that how subjects evaluate others' suffering relative to their own suffering underlies the relationship between IH and hyperaltruistic preferences.

It might be argued that the decision context effect on hyperaltruistic preference was due to loss aversion, a phenomenon that people weigh prospective loss more prominently than monetary gain (*Kahneman and Tversky, 1979*). Our results, however, showed that the relative evaluation of $\Delta m$ did not differ significantly across decision contexts (gain vs. loss, *Figures 2D and 5B*). Furthermore, there was no significant interaction effect of IH and decision contexts on the relative evaluation of $\Delta m$ (*Figure 3D*, *Figure 5—figure supplement 3A*), excluding the role of loss aversion in mediating the context effect on hyperaltruistic preference. From a theoretical point of view, loss aversion only proportionally shifts harm aversion parameter κ (representing the relative importance of $\Delta m$ and $\Delta s$) in the self- and other-conditions, respectively, and will not change the direction of hyperaltruistic preference ($\kappa_{other}$-$\kappa_{self}$).

Instead, we propose that the moral framing of the decision context as helping or harming others may drive subjects' hyperaltruistic preference. Indeed, we showed that subjects were less likely to perceive the task as harming others for monetary benefit when the decision context switched from gain to loss condition (*Figure 6A*). In addition, subjects' harm framing reports fully mediated the correlation between IH and the relative harm sensitivity in the gain conditions (*Figures 5C and 6C*). Decision context exerted its effect by modulating the mediation effect of harm framing report such that the loss context eliminated the correlation between the relative harm sensitivity and IH (*Figure 5C*). Our results are consistent with a recent study where the exogenous moral framing of the task (harming or helping others) significantly biased subjects' prosocial behavior (*Liu et al., 2020*; *Yang et al., 2022*). Subjects behaved more prosocially under the harming frame compared to the helping frame. In our experiments, the decision contexts prompted different endogenous moral framing of the task such that more harm framing led to higher hyperaltruism levels. These results taken together suggest that moral framing may be a critical factor influencing subjects' prosocial preference. Another recent study reported that human subjects can be both egoistic and hyperaltruistic in moral decisions (*Volz et al., 2017*). While the hyperaltruistic preference was elicited by comparing how subjects evaluated others' suffering to their own suffering in the standard monetary gain-pain trade-off task, egoistic tendencies emerged when subjects had to decide whether to harm themselves for others' benefit (compared to harming themselves for their own benefit). These ostensibly puzzling results can be reconciled under the framework of internal moral framing of the task. In order for participants to express moral or prosocial choices, they have to identify how salient a moral code is in place. Internally framing the task as benefiting from others' suffering would be more likely to discourage people from actions they would have otherwise taken if the task is perceived as sacrificing self-interest in order to help others. The personality trait instrument harm (IH) attitudes tap into subjects' relative harm sensitivities, and the correlation between IH and relative harm sensitivity is modulated by the decision context.

In our pre-registered study 2, we directly tested the potential mediating role of moral framing and found that the administration of oxytocin significantly increased subjects' moral framing of the decision context as harming others in both the gain and loss contexts (*Figure 6A*). It is worth noting that oxytocin administration did not affect subjects' IH disposition (*Figure 4—figure supplement 1B*). Therefore, oxytocin effectively nullifies the modulation effects of decision context and preserves the correlation between the relative harm sensitivity and IH in both decision contexts (*Figures 5D and 6C*). Oxytocin has long been featured in the general approach-avoidance hypothesis (*Harari-Dahan and Bernstein, 2014*), which proposes that oxytocin attenuates people's avoidance of negative social or nonsocial stimuli (*Evans et al., 2010*; *Harari-Dahan and Bernstein, 2014*; *Harari-Dahan and Bernstein, 2017*; *Wang and Ma, 2020*). Other studies conducted on both animals and humans suggest that oxytocin increases pain tolerance and attenuates acute pain experience (*Rash et al., 2014*; *Boll et al., 2018*; *Lopes and de L Osório, 2023*). However, hyperaltruistic preference is tied to the differential evaluation of other's suffering relative to subjects' own suffering and thus a general approach-avoidance theory of oxytocin might not be capable of accounting for the decision context-specific effect. Recently, oxytocin has been shown to play a significant role in regulating parochial altruism by promoting both in-group cooperation and out-group aggression (*Zhang et al., 2019*; *De Dreu et al., 2010*; *De Dreu et al., 2020*). Our results corroborated this line of research and highlighted the role of oxytocin in modulating the moral perception of the decision environment. Importantly, we demonstrated that moral perception of the task structure mediated the correlation between subjects' personality attitudes toward harming others (IH) and their relative harm sensitivities, which directly contributed to the emergence of hyperaltruistic preference. Through this lens, decision context can be viewed as the implicit proxy of the mediator (task moral perception). As oxytocin increases the task perception as more harming others for self-interest (*Figure 6A*), the correlation between IH and the relative harm sensitivities is restored.

In summary, in two studies, we demonstrate subjects' hyperaltruistic preference is closely associated with the decision context. Subjects' internal moral framing of the decision context mediates the correlation between personality traits such as IH attitude and the relative harm sensitivity, and the decision context exerts a modulation effect on the mediation effect. Intranasal oxytocin administration drives subjects to be more harm frame oriented and abolishes the modulation effects of the decision contexts. Therefore, our study provides valuable insights into the psychological and cognitive mechanisms underlying hyperaltruistic behavior and highlights the crucial role of oxytocin in shaping

moral decision-making. Finally, our findings carry significant implications for future research on moral behavior and its impairment in specific crisis contexts.

## Materials and methods

### Participants

All subjects were students recruited through an online university platform with written informed consent. Subjects were free of historic or current neurological or psychological disorders and with corrected normal vision. This study was approved by the institutional review board of the school of psychological and cognitive sciences at Peking University.

We conducted a power analysis (G*Power 3.1) to determine the number of subjects sufficient to detect a reliable hyperaltruistic effect reported in the previous literature (*Crockett et al., 2014*; *Crockett et al., 2015*; *Crockett et al., 2017*; *Volz et al., 2017*; *Faul et al., 2009*). Based on the small to medium effect size (Cohen's $d = 0.2$), 75 subjects were needed to detect a significant effect ($\alpha = 0.05$, $\beta = 0.9$, two-by-two within ANOVA) for study 1. We ended up recruiting 83 right-handed subjects for study 1. Two subjects were excluded from data analysis due to their exclusive selection of the same option (more or less painful) across all trials, and 1 subject did not complete the experiment, leaving a total of 80 subjects (40 males, mean age = 21.38±2.67 years). Study 2 (the oxytocin study) was specifically designed to test the oxytocin effect on hyperaltruism and was preregistered (https://osf.io/fhwa9) via the Open Science Framework. Due to the potential confounds of the oxytocin effects, we only recruited male subjects for study 2 (*Lynn et al., 2014*; *Hoge et al., 2014*). It should be noted that in preregistration we originally planned to recruit 60 male subjects for study 2 but ended up recruiting 46 male subjects (mean age = 21.74±2.33 years) based on the sample size reported in previous oxytocin studies (*Liu et al., 2019*; *Wang and Ma, 2020*). Additionally, a power analysis suggested that the sample size >44 should be enough to detect a small to median effect size of oxytocin (Cohen's $d = 0.21$, $\alpha = 0.05$, $\beta = 0.8$) using a 2 × 2 × 2 within-subject design (*Walum et al., 2016*). All the subjects were instructed to avoid taking caffeine, cigarettes, and alcohol 24 h before the experiment and to refrain from eating or drinking 2 h before the experiment.

### Experimental procedures

Each time, two subjects (a pair) arrived with a 5 min interval and entered into separate testing rooms without seeing each other to ensure complete anonymity. After signing the informed consent, subjects completed the pain calibration procedure (described below) in separate rooms. We then followed the protocol developed in the previous literature to assign experimental roles to both subjects before the main task (*Crockett et al., 2014*). Briefly, each subject was told that there was a second subject in the next testing room (whom they would not meet in person). Subjects were instructed that a random coin toss would decide their roles as the decider or receiver in the upcoming money-pain trade-off task. Unbeknownst to the subjects, both of them were assigned to the role of deciders.

For study 2, the experimental procedures were similar to those in study 1. In addition, both subjects were administered with either oxytocin or placebo in two sessions (sessions 1 and 2) of 5–7 days apart in a randomized order, yielding 11 pairs (22 subjects) having the oxytocin session first and the other 12 pairs (24 subjects) taking the placebo session first. Also, once the subjects' task roles (deciders) were announced in session 1, their roles remained in session 2.

### Pain calibration

We adopted a standard pain titration paradigm that has been widely used in previous literature (*Crockett et al., 2014*; *Vlaev et al., 2009*). Electric shocks were generated with a Digitimer DS5 electric stimulator (Digitimer, UK) and applied to the inner side of the subject's left wrist via two electrodes. By slowly increasing or decreasing the electric shock intensities, we asked subjects to rate their pain experience on an 11-point scale ranging from 0 (no pain at all) to 10 (intolerable) until a rating of 10 was reached (subject's maximum tolerance threshold). Next, we generated shocks ranging from 30% to 90% of the subjects' maximum threshold in 10% increments. Each shock intensity was delivered three times. Subjects in total received 21 shocks in randomized order and gave pain ratings on the 11-point scale. For each subject, we fitted a sigmoid function to subjects' subjective pain ratings and chose the current intensity that corresponded to each subject's subjective rating of level 7 from

the derived function (*Figure 1B*). This individualized intensity was then used in the following money-pain trade-off task.

## Oxytocin/placebo administration

The oxytocin and placebo administration procedure was similar to that used in the previous studies (*Ma et al., 2016*; *Liu et al., 2019*). For each treatment, oxytocin or placebo (saline) spray was administered to subjects thrice, consisting of one inhalation of 4 international units (IU) into each nostril resulting in a total of 24 IU. The order of oxytocin and placebo sessions was counterbalanced across subjects. After each session, subjects were asked to rate on a 5-point scale about their perception of the administered substance being oxytocin, with 1 indicating it was unlikely to be oxytocin, 5 indicating strong confidence that it was oxytocin. Subjects' ratings indicated that they had no bias toward the substance used in each session (placebo session: mean = 3.13 ± 0.89; oxytocin session: mean = 3.11 ± 1.16; the difference between the two sessions: $t_{45} = 0.0965$, $P = 0.924$). No subjects reported any side effects after the experiments.

## Experimental design

Study 1 adopted a 2 (decision context: gain vs. loss) × 2(shock recipient: self vs. other) within-subject design (*Figure 1A*). Each subject had to choose between two options representing the trade-off between money and shock. In the gain context, subjects decided whether to inflict more pain (more electric shocks) either on themselves (gain-self condition) or on the receivers (gain-other condition) to gain more money. In the loss context, subjects would decide whether to inflict more pain either on themselves (loss-self condition) or on the receivers (loss-other condition) to avoid a bigger monetary loss. The sequence of the gain and loss contexts was block designed and counterbalanced across subjects. In study 2, we employed a 2 (decision context: gain vs. loss) × 2 (shock recipient: self vs. other) × 2 (treatment: placebo vs. oxytocin) within-subject design to examine oxytocin's effect on subjects' hyperaltruistic preference. Each subject performed the same money-pain trade-off tasks (same as study 1) twice. Approximately 35 min before each experiment and immediately after the pain calibration procedure, each subject was intranasally administered 24 IU oxytocin or placebo (saline) (*Figure 1B*).

In the money-pain trade-off task, subjects were asked to choose between options associated with different levels of electric shocks and monetary amounts. Crucially, more painful options were always associated with larger monetary gain (or smaller monetary loss), while less painful options offered smaller monetary gains (or larger monetary losses). The experimental task was coded with Psychtoolbox (version 3.0.17) in MATLAB (2018b). We first created 60 gain trials using the procedure reported in earlier research (*Crockett et al., 2014*). Specifically, each trial was determined by a combination of the differences of shocks ($\Delta s$, ranging from 1 to 19, with increment of 1) and money ($\Delta m$, ranging from ¥0.2 to ¥19.8, with increment of ¥0.2) between the two options, resulting in a total of 19×99 = 1881 pairs of [$\Delta s$, $\Delta m$]. To ensure the trials were suitable for most subjects, we evenly distributed the desired ratio $\Delta m/(\Delta s + \Delta m)$ between 0.01 and 0.99 across 60 trials for each condition. For each trial, we selected the closest [$\Delta s$, $\Delta m$] pair from the [$\Delta s$, $\Delta m$] pool to the specific $\Delta m/(\Delta s + \Delta m)$ ratio, which was then used to determine the actual money and shock amounts of two options. The shock amount ($S_{less}$) for the less painful option was an integer drawn from the discrete uniform distribution [1~19], constrained by $S_{less}+\Delta s<20$. Similarly, the money amount ($M_{less}$) for the less painful option was drawn from a discrete uniform distribution [¥0.2 ~ ¥19.8], with the constraint of $M_{less} + \Delta m < 20$. Once the $S_{less}$ and $M_{less}$ were selected, the shock ($S_{more}$) and money ($M_{more}$) magnitudes for the more painful option were calculated as: $S_{more} = S_{less} + \Delta s$, $M_{more} = M_{less} + \Delta m$. For the loss condition, we flipped the signs for the monetary amount and then switched the monetary amounts between the more and less painful options. For example, the two options [¥15 and 10 shocks; ¥10 and 5 shocks] in the gain condition would turn to [- ¥10 and 10 shocks; - ¥15 and 5 shocks] in the loss condition. Subjects completed 60 trials each for all the four decision context and shock-recipient combinations (gain-self, gain-other, loss-self, and loss-other), thus yielding a total of 240 trials delivered across two blocks (gain and loss blocks). We randomly generated each subject's trial set using the above method. Across trials, $\Delta s$ and $\Delta m$ were uncorrelated (r= -0.0012,P= 0.986). The trial sequences and the more/less painful option location (left vs. right) on the computer screen were randomized across subjects within each block.

For each trial, subjects had a maximum of 6 s to choose either the more or less painful option by pressing a keyboard button with their right or left index finger. Button presses resulted in the chosen option being highlighted on the screen for 1 s. If no choice was entered during the 6 s response window, a message 'Please respond faster!' was displayed for 1 s, and this trial was repeated. An inter-trial interval (ITI) of 1 s was introduced before the beginning of the next trial (*Figure 1B*). Each subject was endowed with ¥40 at the beginning of the task (the maximum amount subjects could have lost in each loss-self and loss-other trials together). At the end of the experiment, one trial for each experimental condition was randomly selected and subjects were renumerated and shocked according to the combined payoff (four conditions) and electric shocks (only in gain-self and loss-self conditions). Each subject received a ¥60 show-up fee in study 1 (¥220 in study 2), and the average total subject fee is ¥97 (¥318 in study 2).

## Personality trait measures and questionnaires

The Oxford utilitarianism scale was utilized to evaluate two independent dimensions that drive people's prosocial preference in a moral dilemma (*Kahane et al., 2018*): the dimension of IH measures individuals' willingness to compromise moral principles by either breaking rules or causing harm to others to achieve favorable outcomes, whereas the IB captures their levels of impartial concern for the well-being of the general population. IH has four items including statements such as "Sometimes it is morally necessary for innocent people to die as collateral damage if more people are saved overall". The IB measure contains 5 items such as "It is morally wrong to keep money that one doesn't really need if one can donate it to causes that provide effective help to those who will benefit a great deal". Subjects responded to each IH and IB item via a 7-point Likert scale. IH and IB scores were derived by calculating the average score of all items on the subscale for each subject. We also measured subjects' empathetic concern (EC), the identification with the whole of humanity and concern for future generation, using the interpersonal reactivity index (IRI) (*Davis, 2011*). The EC comprises seven items with a 5-point rating scale and has been shown to be correlated with both IH and IB (*Figure 3—figure supplement 1*). Therefore, we performed multiple regression analysis to examine the relationships between IB, IH, and moral behaviors, with EC included as a covariate of no interest (see *Tables 1 and 2* for details).

All subjects completed debriefing questionnaires that assessed their beliefs about the experimental setup, such as whether they believed that the recipient would actually receive the electric shocks. Moreover, for study 2, we also collected another questionnaire by asking subjects how they perceived the task structure (whether they regarded the experiment as helping or harming other subjects) in both decision contexts (gain and loss). Subjects rated their moral perception within gain and loss contexts using a scale ranging from –4 to 4. Positive values indicated harm frame (inflicting pain on the receiver to increase monetary gain/avoid monetary loss), whereas negative values indicated help frame (sacrificing greater monetary gain/accepting greater monetary loss to alleviate others' pain) with 0 indicating neutrality. Therefore, a larger rating indicated the subject's moral perception of a more harming frame.

## Data analysis

### Harm aversion model

We analyzed subjects' choice data using the harm aversion model (*Crockett et al., 2017*), where choices were driven by the subjective value difference ($\Delta V$) between the less and more painful options (*Equation 1*). Parameter κ ($0 \leq \kappa \leq 1$ quantifies the relative weight deciders attribute to electric shock versus money.

$$\Delta V = (1 - \kappa)\Delta m - \kappa\Delta s \tag{1}$$

We separately estimated κ in the four conditions related to both the shock recipients (self vs. other) and decision contexts (gain vs. loss) and thus yielded $\kappa_{gain-self}$, $\kappa_{gain-other}$, $\kappa_{loss-self}$, and $\kappa_{loss-other}$, respectively. $\Delta m$ and $\Delta s$ denoted the objective differences in money and electric shocks between the more and less painful options. In this model, trial-wise $\Delta V$ was transformed into choice probability via a softmax function where γ serves as a subject-specific parameter for choice consistency (*Equation 2*) (see *Figure 2—figure supplement 2* for the model comparison results that confirmed a single γ for all conditions fitted subjects' behavior better than other candidate models).

$$P\,(\text{less painful option}) = \frac{1}{1 + e^{-\gamma \Delta V}} \tag{2}$$

Model parameters were estimated for each subject using maximum likelihood estimation (MLE), and the maximization was achieved using the fminsearchbnd function in MATLAB (MathWorks). We repeated the estimation with 300 random initial parameter values to achieve stable parameter estimators.

## Regression models

We applied the mixed-effect logistic regression model to investigate how $\Delta m$ and $\Delta s$ independently influenced subjects' choices. Although this regression model seems equivalent to the harm aversion model mentioned above, the regression coefficients obtained from the regression model allowed us to separately examine how the decision contexts and oxytocin treatment affected subjects' sensitivities toward $\Delta m$ and $\Delta s$. In the regression model 1 of study 1, choices were coded as 1 if subjects chose the less painful option and 0 otherwise. Additionally, decision context (1 for gain and 0 for loss) and shock recipient (1 for self and 0 for other) were treated as categorical variables. Each regressor had a fixed effect across all subjects and a random effect specific to each subject. The regression model 1 was specified as follows (*Wilkinson and Rogers, 1973*):

$$
\begin{aligned}
choice \sim \quad & \Delta s * context * recipient + \Delta m * context * recipient + \\
& (1 + \Delta s * context * recipient + \Delta m * context * recipient \mid subject)
\end{aligned}
$$

To examine how oxytocin affected subjects' behavior, in study 2, we expanded regression model 1 by including the variable of treatment (1 for oxytocin treatment and 0 for placebo) as an additional categorical independent variable. The regression model 2 was described as

$$
\begin{aligned}
choice \sim \quad & \Delta s * context * recipient * treatment + \Delta m * context * recipient * treatment + \\
& (1 + \Delta s * context * recipient * treatment + \Delta m * context * recipient * treatment \mid subject)
\end{aligned}
$$

## Moderated mediation analysis

We set out to examine whether subjects' internal moral framing mediates the association between their personality traits (IH) and moral preferences and, more importantly, whether the mediation is context (gain vs. loss) specific. To test the moderated mediation, we constructed a mediation model where the mediator (M):

$$M \sim \beta_0^M + \beta_1^M IH + \beta_2^M DC + \beta_3^M IH \times DC + \beta_4^M IB + \beta_5^M EC\,(modelM)$$

and the dependent variable model was

$$Y \sim \beta_0^Y + \beta_1^Y IH + \beta_2^Y M + \beta_3^Y DC + \beta_4^Y IH \times DC + \beta_5^Y M \times DC + \beta_6^Y IB + \beta_7^Y EC\,(modelY)$$

where *Y, M* respectively referred to the dependent variable (relative harm sensitivity) and the mediator (harm framing report). Variables IH, DC, IB, and EC corresponded to independent variable IH, decision context (gain vs. loss, the moderator variable), IB (variable of no interest), and empathic concern (EC, variable of no interest), accordingly. The existence of an interaction effect ($\beta_3^M, \beta_4^Y, \beta_5^Y$) in either path a (X→M), path b (M→Y), or path c' (X→Y) in the model is sufficient to claim moderation of mediation (*Preacher and Hayes, 2008*).

We utilized the bootstrapping procedure with 5000 bootstrap resamples to analyze the data (*Preacher et al., 2007*; *Hayes, 2015*). This analysis was conducted separately for the placebo and oxytocin sessions. Our results showed that the interaction coefficients (placebo: $\beta_4^Y = 0.036 \pm 0.029, t_{84} = 1.267, P = 0.209$; $\beta_5^Y = 0.0242 \pm 0.017, t_{84} = 1.415, P = 0.161$; oxytocin: $\beta_4^Y = 0.006 \pm 0.010, t_{84} = 0.319, P = 0.751$; $\beta_5^Y = 0.003 \pm 0.012, t_{84} = 0.246, P_= 0.806$) in model Y were not significant, while $\beta_3^M$ was significant in the placebo session ($\beta_3^M = 0.784 \pm 0.343, t_{86} = 2.288, P = 0.025$) but not the oxytocin session ($\beta_3^M = -0.137 \pm 0.291, t_{86} = -0.472, P = 0.638$), indicating that the moderation effect was present only on the path a (see *Figure 6B*) in the placebo condition. This aligns with our findings that decision contexts modulated the correlation between IH and relative harm

sensitivities (*Figures 3C and 5C*). Therefore, we report all the results from the reduced version of the moderated mediation model (model $Y'$, see *Table 3* for details).

$$Y' \sim \beta_0^{Y'} + \beta_1^{Y'} IH + \beta_2^{Y'} M + \beta_3^{Y'} DC + \beta_6^{Y'} IB + \beta_7^{Y'} EC \,(\text{model Y'})$$

### Statistics and software

ANOVAs and non-parametric analyses were conducted with SPSS 27.0, whereas regression analyses were performed using 'fitglme' and 'fitlm' functions in MATLAB (2022b). The moderated mediation analysis was performed using 'bruceR' and 'mediation' packages in R (version 4.2.2). All the reported *P*-values are two-tailed.

## Acknowledgements

This work was supported by the National Science and Technology Innovation 2030 Major Program (2021ZD0203702), National Natural Science Foundation of China Grants (32071090) to J.L.

## Additional information

### Funding

| Funder | Grant reference number | Author |
| --- | --- | --- |
| National Science and Technology Major Project | 2021ZD0203702 | Jian Li |
| National Natural Science Foundation of China | 32441111 | Jian Li |

The funders had no role in study design, data collection and interpretation, or the decision to submit the work for publication.

### Author contributions

Hong Zhang, Conceptualization, Formal analysis, Investigation, Visualization, Writing - original draft; Yinmei Ni, Software, Formal analysis, Supervision, Methodology, Writing - review and editing; Jian Li, Formal analysis, Supervision, Funding acquisition, Project administration, Writing - review and editing

### Author ORCIDs

Yinmei Ni (ID) https://orcid.org/0000-0003-3614-1828
Jian Li (ID) https://orcid.org/0000-0002-3941-2622

### Ethics

Human subjects: All subjects were students recruited through online university platform with written informed consent. This study was approved by the institutional review board of the school of psychological and cognitive sciences at Peking University.

Reviewer #1 (Public review): https://doi.org/10.7554/eLife.102756.3.sa1
Reviewer #2 (Public review): https://doi.org/10.7554/eLife.102756.3.sa2
Reviewer #3 (Public review): https://doi.org/10.7554/eLife.102756.3.sa3
Author response https://doi.org/10.7554/eLife.102756.3.sa4

## Additional files

### Supplementary files
MDAR checklist

### Data availability
The data and analysis code that support the findings of this study are available at https://osf.io/tpfeg/.

The following dataset was generated:

| Author(s) | Year | Dataset title | Dataset URL | Database and Identifier |
|---|---|---|---|---|
| Zhang H | 2023 | Oxytocin salvages context-specific hyperaltruistic preference | https://osf.io/tpfeg/ | Open Science Framework, tpfeg |

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
