## [Editor Report · eLife Assessment]

This revised paper provides **valuable** findings that altruistic tendency during moral decision-making is gain/loss context-dependent and oxytocin can restore the absence of altruistic choices in the loss domain. The methods and analyses are **solid**, yet the study could still benefit from better overall framing and more clarity and precision in the definition of key constructs, as pointed out by reviewers. If these concerns are addressed, this study would be of interest to social scientists and neuroscientists who work on moral decision-making and oxytocin.

---

## [Referee Report · Reviewer #1 (Public review)]

Summary:

Zhang et al. addressed the question of whether hyperaltruistic preference is modulated by decision context and tested how oxytocin (OXT) may modulate this process. Using an adapted version of a previously well-established moral decision-making task, healthy human participants in this study undergo decisions that gain more (or lose less, termed as context) meanwhile inducing more painful shocks to either themselves or another person (recipient). The alternative choice is always less gain (or more loss) meanwhile less pain. Through a series of regression analyses, the authors reported that hyperaltruistic preference can only be found in the gain context but not in the loss context, however, OXT reestablished the hyperaltruistic preference in the loss context similar to that in the gain context.

Strengths:

This is a solid study that directly adapted a previously well-established task and the analytical pipeline to assess hyperaltruistic preference in separate decision contexts. Context-dependent decisions have gained more and more attention in literature in recent years, hence this study is timely. It also links individual traits (via questionnaires) with task performance, to test potential individual differences. The OXT study is done with great methodological rigor, including pre-registration. Both studies have proper power analysis to determine the sample size.

Weaknesses:

Despite the strengths, multiple analytical decisions have to be explained, justified, or clarified. Also, there is scope to enhance the clarity and coherence of the writing - as it stands, readers will have to go back and forth to search for information. Last, it would be helpful to add line numbers in the manuscript during the revision, as this will help all reviewers to locate the parts we are talking about.

Introduction:

(1) The introduction is somewhat unmotivated, with key terms/concepts left unexplained until relatively late in the manuscript. One of the main focuses in this work is "hyperaltruistic", but how is this defined? It seems that the authors take the meaning of "willing to pay more to reduce other's pain than their own pain", but is this what the task is measuring? Did participants ever need to PAY something to reduce the other's pain? Note that some previous studies indeed allow participants to pay something to reduce other's pain. And what makes it "HYPER-altruistic" rather than simply "altruistic"? Plus, in the intro, the authors mentioned that the "boundary conditions" remain unexplored, but this idea is never touched again. What do boundary conditions mean here in this task? How do the results/data help with finding out the boundary conditions? Can this be discussed within wider literature in the Discussion section? Last, what motivated the authors to examine decision context? It comes somewhat out of the blue that the opening paragraph states that "We set out to [...] decision context", but why? Are there other important factors? Why decision context is more important than studying those others?

Experimental design:

(2) The experiment per se is largely solid, as it followed a previously well-established protocol. But I am curious about how the participants got instructed? Did the experimenter ever mention the word "help" or "harm" to the participants? It would be helpful to include the exact instructions in the SI.

(3) Relatedly, the experimental details were not quite comprehensive in the main text. Indeed, Methods come after the main text, but to be able to guide readers to understand what was going on, it would be very helpful if the authors could include some necessary experimental details at the beginning of the Results section.

Statistical analysis

(3) One of the main analyses uses the harm aversion model (Eq1) and the results section keeps referring to one of the key parameters of it (ie, k). However, it is difficult to understand the text without going to the Methods section below. Hence it would be very helpful to repeat the equation also in the main text. A similar idea goes to the delta_m and delta_s terms - it will be very helpful to give a clear meaning of them, as nearly all analyses rely on knowing what they mean.

(4) There is one additional parameter gamma (choice consistency) in the model. Did the authors also examine the task-related difference of gamma? This might be important as some studies have shown that the other-oriented choice consistency may differ in different prosocial contexts.

(5) I am not fully convinced that the authors included two types of models: the harm aversion model and logistic regression models. Indeed, the models look similar, and the authors have acknowledged that. But I wonder if there is a way to combine them? For example:

Choice ~ delta_V * context * recipient (*Oxt_v._placebo)

The calculation of delta_V follows Equation 1.

Or the conceptual question is, if the authors were interested in the specific and independent contribution of dalta_m and dalta_s to behavior, as their logistic model did, why the authors examine the harm aversion first, where a parameter k is controlling for the trade-off? One way to find it out is to properly run different models and run model comparison. In the end, it would be beneficial to only focus on the "winning" model to draw inferences.

(6) The interpretation of the main OXT results needs to be more cautious. According to the operationalization, "hyperaltruistic" is the reduction of pain of others (higher % of choosing the less painful option) relative to the self. But relative to the placebo (as baseline), OXT did not increase the % of choosing the less painful option for others, rather, it decreased the % of choosing the less painful option for themselves. In other words, the degree of reducing other's pain is the same under OXT and placebo, but the degree of benefiting self-interest is reduced under OXT. I think this needs to be unpacked, and some of the wording needs to be changed. I am not very familiar with the OXT literature, but I believe it is very important to differentiate whether OXT is doing something on self-oriented actions vs other-oriented actions. Relatedly, for results such as that in Fig5A, it would be helpful to not only look at the difference, but also the actual magnitude of the sensitivity to the shocks, for self and others, under OXT and placebo.

Comments on revisions:

I did not change my original public review, as I think it can still be helpful for the field to see the reasoning and argument.

For the revision, the authors have done a thorough job of addressing my previous comments and questions.

The only aspect I would like to ask is that, it would still be great to have a clear definition of hyperaltruism. As it stands, hyperaltruism refers to "people's willingness to pay more to reduce other's pain than

their own pain", ie, this means the "hyper" bit is considered with respect to "self". But shouldn't hyperaltruism be classified contrasting "normal" altruism?

It is fine that it follows a previously published work (Crockett et al., 2014), but it would still be necessary to explain/define the construct being tested in a standalone fashion rather than letting readers to go back to the original work.

---

## [Referee Report · Reviewer #2 (Public review)]

Summary:

In this manuscript, the authors reported two studies where they investigated the context effect of hyperaltruistic tendency in moral decision-making. They replicated the hyperaltruistic moral preference in the gain domain, where participants inflicted electric shocks to themselves or another person in exchange for monetary profits for themselves. In the loss domain, such hyperaltruistic tendency abolished. Interestingly, oxytocin administration reinstated the hyperaltruistic tendency in the loss domain. The authors also examined the correlation between individual differences in utilitarian psychology and the context effect of hyperaltruistic tendency.

Strengths:

(1) The research question - the boundary condition of hyperaltruistic tendency in moral decision-making and its neural basis - is theoretically important.

(2) Manipulating the brain via pharmacological means offers causal understanding of the neurobiological basis of the psychological phenomenon in question.

(3) Individual difference analysis reveals interesting moderators of the behavioral tendency.

Weaknesses:

(1) The theoretical hypothesis needs to be better justified. There are studies addressing the neurobiological mechanism of hyperaltruistic tendency, which the authors unfortunately skipped entirely.

(2) There are some important inconsistencies between the preregistration and the actual data collection/analysis, which the authors did not justify.

(3) Some of the exploratory analysis seems underpowered (e.g., large multiple regression models with only about 40 participants).

(4) Inaccurate conceptualization of utilitarian psychology and the questionnaire used to measure it.

Comments on revisions:

The authors have addressed the weakness in the second round of revision

---

## [Referee Report · Reviewer #3 (Public review)]

Summary:

In this study, the authors aimed to index individual variation in decision-making when decisions pit the interests of the self (gains in money, potential for electric shock) against the interests of an unknown stranger in another room (potential for unknown shock). In addition, the authors conducted an additional study in which male participants were either administered intranasal oxytocin or placebo before completing the task to identify the role of oxytocin in moderating task responses. Participants' choice data was analyzed using a harm aversion model in which choices were driven by the subjective value difference between the less and more painful options.

Strengths:

Overall, I think this is a well-conducted, interesting, and novel set of research studies exploring decision-making that balances outcomes for the self versus a stranger, and the potential role of the hormone oxytocin (OT) in shaping these decisions. The pain component of the paradigm is well designed, as is the decision-making task, and overall the analyses were well suited to evaluating and interpreting the data. Advantages of the task design include the absence of deception, e.g., the use of a real study partner and real stakes, as a trial from the task was selected at random after the study and the choice the participant made were actually executed.

Weaknesses:

The primary weakness of the paper concerns its framing. Although it purports to be measuring "hyper-altruism," which is the same term used in prior similar (although not identical) designs, I do not believe the task constitutes altruism, but rather the decision to engage, or not engage, in instrumental aggression.

I continue to believe that when in the "other" trials the only outcome possible for the study partner is pain, and the only outcome possible for the participant is monetary gain, these trials measure decisions about instrumental aggression. That is the exact definition of instrumental aggression is: causing others harm for personal gain. Altruism is not equivalent to refraining from engaging in instrumental aggression, although some similar mechanisms may support both. True altruism would be to accept shocks to the self for the other's benefit (e.g., money). The interpretation of this task as assessing instrumental aggression is supported by the fact that only the Instrumental Harm subscale of the OUS was associated with outcomes in the task, but not the Impartial Benevolence subscale. By contrast, the IB subscale is the one more consistently associated with altruism (e.g,. Kahane et al 2018; Amormino at al, 2022) I believe it is important for scientific accuracy for the paper, including the title, to be rewritten to reflect what it is testing.

Although I recognize similar tasks have been previously characterized as "hyper-altruism" I do not believe that is sufficient justification for continuing to promulgate this descriptor without any caveats. I hope the authors will engage more seriously with the idea that this is what the task is measuring.

Relatedly, in the introduction, I believe it would be important to discuss the non-symmetry of moral obligations related to help/harm--we have obligations not to harm strangers but no obligation to help strangers. This is another reason I do not think the term "hyper altruism" is a good description for this task--given it is typically viewed as morally obligatory not to harm strangers, choosing not to harm them is not "hyper" altruistic (and again, I do not view it as obviously altruism at all).

---

## [Author Response]

The following is the authors’ response to the original reviews

**Reviewer #1:**
Despite the strengths, multiple analytical decisions have to be explained, justified, or clarified. Also, there is scope to enhance the clarity and coherence of the writing - as it stands, readers will have to go back and forth to search for information. Last, it would be helpful to add line numbers in the manuscript during the revision, as this will help all reviewers to locate the parts we are talking about.

We thank the reviewer’s suggestions have added the line numbers to the revised manuscript.

(1) Introduction:The introduction is somewhat unmotivated, with key terms/concepts left unexplained until relatively late in the manuscript. One of the main focuses in this work is "hyperaltruistic", but how is this defined? It seems that the authors take the meaning of "willing to pay more to reduce other's pain than their own pain", but is this what the task is measuring? Did participants ever need to PAY something to reduce the other's pain? Note that some previous studies indeed allow participants to pay something to reduce other's pain. And what makes it "HYPER-altruistic" rather than simply "altruistic"?

As the reviewer noted, we adopted a well-established experimental paradigm to study the context-dependent effect on hyper-altruism. Altruism refers to the fact that people take others’ welfare into account when making decisions that concern both parties. Research paradigms investigating altruistic behavior typically use a social decision task that requires participants to choose between options where their own financial interests are pitted against the welfare of others (FeldmanHall et al., 2015; Hu et al., 2021; Hutcherson et al., 2015; Teoh et al., 2020; Xiong et al., 2020). On the other hand, the hyperaltruistic tendency emphasizes subjects’ higher valuation to other’s pain than their own pain (Crockett et al., 2014, 2015, 2017; Volz et al., 2017). One example for the manifestation of hyperaltruism would be the following scenario: the subject is willing to forgo $2 to reduce others’ pain by 1 unit (social-decision task) and only willing to forgo $1 to reduce the same amount of his/her own pain (self-decision task) (Crockett et al., 2014). On the contrary, if the subjects are willing to forgo less money to reduce others’ suffering in the social decision task than in the self-decision task, then it can be claimed that no hyperaltruism is observed. Therefore, hyperaltruistic preference can only be measured by collecting subjects’ choices in both the self and social decision tasks and comparing the choices in both tasks.

In our task, as in the studies before ours (Crockett et al., 2014, 2015, 2017; Volz et al., 2017), subjects in each trial were faced with two options with different levels of pain on others and monetary payoffs on themselves. Based on subjects’ choice data, we can infer how much subjects were willing to trade 1 unit of monetary payoff in exchange of reducing others’ pain through the regression analysis (see Figure 1 and methods for the experimental details). We have rewritten the introduction and methods sections to make this point clearer to the audience.

Plus, in the intro, the authors mentioned that the "boundary conditions" remain unexplored, but this idea is never touched again. What do boundary conditions mean here in this task? How do the results/data help with finding out the boundary conditions? Can this be discussed within wider literature in the Discussion section?

Boundary conditions here specifically refer to the variables or decision contexts that determine whether hyperaltruistic behavior can be elicited. Individual personality trait, motivation and social relationship may all be boundary conditions affecting the emergence of hyperaltruistic behavior. In our task, we specifically focused on the valence of the decision context (gain vs. loss) since previous studies only tested the hyperaltruistic preference in the gain context and the introduction of the loss context might bias subjects’ hyperaltruistic behavior through implicit moral framing.

We have explained the boundary conditions in the revised introduction (Lines 45 ~ 49).

“However, moral norm is also context dependent: vandalism is clearly against social and moral norms yet vandalism for self-defense is more likely to be ethically and legally justified (the Doctrine of necessity). Therefore, a crucial step is to understand the boundary conditions for hyperaltruism.”

Last, what motivated the authors to examine the decision context? It comes somewhat out of the blue that the opening paragraph states that "We set out to [...] decision context", but why? Are there other important factors? Why decision context is more important than studying those others?

We thank the reviewer for the comment. The hyperaltruistic preference was originally demonstrated between conditions where subjects’ personal monetary gain was pitted against others’ pain (social-condition) or against subjects’ own suffering (self-condition) (Crockett et al., 2014). Follow up studies found that subjects also exhibited strong egoistic tendencies if instead subjects needed to harm themselves for other’s benefit in the social condition (by flipping the recipients of monetary gain and electric shocks) (Volz et al., 2017). However, these studies have primarily focused on the gain contexts, neglecting the fact that valence could also be an influential factor in biasing subjects’ behavior (difference between gain and loss processing in humans). It is likely that replacing monetary gains with losses in the money-pain trade-off task might bias subjects’ hyperaltruistic preference due to heightened vigilance or negative emotions in the face of potential loss (such as loss aversion) (Kahneman & Tversky, 1979; Liu et al., 2020; Pachur et al., 2018; Tom et al., 2007; Usher & McClelland, 2004; Yechiam & Hochman, 2013). Another possibility is that gain and loss contexts may elicit different subjective moral perceptions (or internal moral framings) in participants, affecting their hyperaltruistic preferences (Liu et al., 2017; Losecaat Vermeer et al., 2020; Markiewicz & Czupryna, 2018; Wu et al., 2018). In our manuscript, we did not strive to compare which factors might be more important in eliciting hyperaltruistic behavior, but rather to demonstrate the crucial role played by the decision context and to show that the internal moral framing could be the mediating factor in driving subjects’ hyperaltruistic behavior. In fact, we speculate that the egoistic tendencies found in the Volz et al. 2017 study was partly driven by the subjects’ failure to engage the proper internal moral framing in the social condition (harm for self, see Volz et al., 2017 for details).

(2) Experimental Design:(2a) The experiment per se is largely solid, as it followed a previously well-established protocol. But I am curious about how the participants got instructed? Did the experimenter ever mention the word "help" or "harm" to the participants? It would be helpful to include the exact instructions in the SI.

In the instructions, we avoided words such as “harm”, “help”, or other terms reminding subjects about the moral judgement of the decisions they were about to make. Instead, we presented the options in a neutral and descriptive manner, focusing only on the relevant components (shocks and money). The instructions for all four conditions are shown in supplementary Fig. 9.

(2b) Relatedly, the experimental details were not quite comprehensive in the main text. Indeed, the Methods come after the main text, but to be able to guide readers to understand what was going on, it would be very helpful if the authors could include some necessary experimental details at the beginning of the Results section.

We thank the reviewer’s suggestion. We have now provided a brief introduction of the experimental details in the revised results section (Lines 125 ~132).

“Prior to the money-pain trade-off task, we individually calibrated each subject’s pain threshold using a standard procedure[4–6]. This allowed us to tailor a moderate electric stimulus that corresponded to each subject’s subjective pain intensity. Subjects then engaged in 240 decision trials (60 trials per condition), acting as the “decider” and trading off between monetary gains or losses for themselves and the pain experienced by either themselves or an anonymous “pain receiver” (gain-self, gain-other, loss-self and loss-other, see Supplementary Fig. 8 for the instructions and also see methods for details).”

(3) Statistical Analysis(3a) One of the main analyses uses the harm aversion model (Eq1) and the results section keeps referring to one of the key parameters of it (ie, k). However, it is difficult to understand the text without going to the Methods section below. Hence it would be very helpful to repeat the equation also in the main text. A similar idea goes to the delta_m and delta_s terms - it will be very helpful to give a clear meaning of them, as nearly all analyses rely on knowing what they mean.

We thank the reviewer’s suggestion. We have now added the equation of the harm aversion model and provided more detailed description to the equations in the main text (Lines 150 ~155).

“We also modeled subjects’ choices using an influential model where subjects’ behavior could be characterized by the harm (electric shock) aversion parameter κ, reflecting the relative weights subjects assigned to ∆m and ∆s, the objective difference in money and shocks between the more and less painful options, respectively (∆V=(1-κ)∆m - κ∆s Eq.1, See Methods for details)[4–6]. Higher κ indicates that higher sensitivity is assigned to ∆s than ∆m and vice versa.”

(3b) There is one additional parameter gamma (choice consistency) in the model. Did the authors also examine the task-related difference of gamma? This might be important as some studies have shown that the other-oriented choice consistency may differ in different prosocial contexts.

To examine the task-related difference of choice consistency (γ), we compared the performance of 4 candidate models:

Model 1 (M1): The choice consistency parameter γ remains constant across shock recipients (self vs. other) and decision contexts (gain vs. loss).

Model 2 (M2): γ differs between the self- and other-recipient conditions, with *γself* and *γother* representing the choice consistency when pain is inflicted on him/her-self or the other-recipient.

Model 3 (M3): γ differs between the gain and loss conditions, with *γgain* and *γloss* representing the choice consistencies in the gain and loss contexts, respectively.

Model 4 (M4): γ varies across four conditions, with *γself-gain*, *γother-gain*, *γself-loss* and *γother-loss* capturing the choice consistency in each condition.

Supplementary Fig. 10 shows, after fitting all the models to subjects’ choice behavioral data, model 1 (M1) performed the best among all the four candidate models in both studies (1 & 2) with the lowest Bayesian Information Criterion (BIC). Therefore, we conclude that factors such as the shock recipients (self vs. other) and decision contexts (gain vs. loss) did not significantly influence subjects’ choice consistency and report model results using the single choice consistency parameter.

(3c) I am not fully convinced that the authors included two types of models: the harm aversion model and the logistic regression models. Indeed, the models look similar, and the authors have acknowledged that. But I wonder if there is a way to combine them? For example:Choice ~ delta_V * context * recipient (*Oxt_v._placebo)The calculation of delta_V follows Equation 1.Or the conceptual question is, if the authors were interested in the specific and independent contribution of dalta_m and dalta_s to behavior, as their logistic model did, why did the authors examine the harm aversion first, where a parameter k is controlling for the trade-off? One way to find it out is to properly run different models and run model comparisons. In the end, it would be beneficial to only focus on the "winning" model to draw inferences.

The reviewer raised an excellent point here. According to the logistic regression model, we have:\begin{document}$$\displaystyle \log \left(\frac{P}{1-P}\right)=\beta_{0}+\beta_{\Delta m} \Delta m+\beta_{\Delta s} \Delta s$$\end{document}

Where *P* is the probability of selecting the less harmful option. Similarly, if we combine Eq.1 (∆V=1-κ) ∆m-κ∆s and Eq.2 \begin{document}$P(less\,pain\, option)=\frac{1}{1+{e}^{-\gamma \Delta V}}$\end{document} of the harm aversion model, we have:\begin{document}$$\displaystyle \log \left(\frac{P}{1-P}\right)=\gamma(1-\mathrm{K}) \Delta \mathrm{m}-\gamma \mathrm{K} \Delta \mathrm{S}$$\end{document}

If we ignore the constant term *β*_0_ from the logistic regression model, the harm aversion model is simply a reparameterization of the logistic regression model. The harm aversion model was implemented first to derive the harm aversion parameter (κ), which is an parameter in the range of [0 1] to quantify how subjects value the relative contribution of Δm and Δs between options in their decision processes. Since previous studies used the term *κother*-*κself* to define the magnitude of hyperaltruistic preference, we adopted similar approach to compare our results with previous research under the same theoretical framework. However, in order to investigate the independent contribution of Δm and Δs, we will have to take γ into account (we can see that the *β∆m* and *β∆s* in the logistic regression model are not necessarily correlated by nature; however, in the harm aversion model the coefficients (1-κ) and κ is always strictly negatively correlated (see Eq. 1). Only after multiplying γ, the correlation between γ(1-κ) and γκ will vary depending on the specific distribution of γ and κ). In summary, we followed the approach of previous research to estimate harm aversion parameter κ to compare our results with previous studies and to capture the relative influence between Δm and Δs. When we studied the contextual effects (gain vs. loss or placebo vs. control) on subjects’ behavior, we further investigated the contextual effect on how subjects evaluated Δm and Δs, respectively. The two models (logistic regression model and harm aversion model) in our study are mathematically the same and are not competitive candidate models. Instead, they represent different aspects from which our data can be examined.

We also compared the harm aversion model with and without the constant term β_0_ in the choice function. Adding a constant term β_0_ the above Equation 2 becomes:\begin{document}$$\displaystyle P(\rm less\, painful\, option)=\frac{1}{1+e^{-\gamma \Delta V+\beta_{0}}}$$\end{document}

As the following figure shows, the hyperaltruistic parameters (*κother*-*κself*) calculated from the harm aversion model with the constant term (panels A & B) have almost identical patterns as the model without the constant term (panels C & D, i.e. Figs. 2B & 4B in the original manuscript) in both studies.

**Author response image 1. sa4fig1:** Figs. 2B & 4B in the original manuscript in both studies.

(3d) The interpretation of the main OXT results needs to be more cautious. According to the operationalization, "hyperaltruistic" is the reduction of pain of others (higher % of choosing the less painful option) relative to the self. But relative to the placebo (as baseline), OXT did not increase the % of choosing the less painful option for others, rather, it decreased the % of choosing the less painful option for themselves. In other words, the degree of reducing other's pain is the same under OXT and placebo, but the degree of benefiting self-interest is reduced under OXT. I think this needs to be unpacked, and some of the wording needs to be changed. I am not very familiar with the OXT literature, but I believe it is very important to differentiate whether OXT is doing something on self-oriented actions vs other-oriented actions. Relatedly, for results such as that in Figure 5A, it would be helpful to not only look at the difference but also the actual magnitude of the sensitivity to the shocks, for self and others, under OXT and placebo.

We thank the reviewer for this thoughtful comment. As the reviewer correctly pointed out, “hyperaltruism” can be defined as “higher % of choosing the less painful option to the others relative to the self”. Closer examination of the results showed that both the degrees of reducing other’s pain as well as reducing their own pain decreased under OXT (Figure 4A). More specifically, our results do not support the claim that “In other words, the degree of reducing others’ pain is the same under OXT and placebo, but the degree of benefiting self-interest is reduced under OXT.” Instead, the results show a significant reduction in the choice of less painful option under OXT treatment for both the self and other conditions (the interaction effect of OXT vs. placebo and self vs. other: *F*_1.45_ = 16.812, *P* < 0.001, *η*^2^ = 0.272, simple effect OXT vs. placebo in the self- condition: *F*_1.45_ = 59.332, *P* < 0.001, *η*^2^ = 0.569, OXT vs. placebo in the other-condition: *F*_1.45_ = 14.626, *P* < 0.001, *η*^2^ = 0.245, repeated ANOVA, see Figure 4A).

We also performed mixed-effect logistic regression analyses where subjects’ choices were regressed against and in different valences (gain vs. loss) and recipients (self vs. other) conditions in both studies 1 & 2 (Supplementary Figs. 1 & 6). As we replot supplementary Fig. 6 and panel B (included as Supplementary Fig. 8 in the supplementary materials) in the above figure, we found a significant treatment × ∆*s* (differences in shock magnitude between the more and less painful options) interaction effect β=0.136±0.029*P* < = 0.001, 95% CI=[-0.192, -0.079]), indicating that subject’s sensitivities towards pain were indeed different between the placebo and OXT treatments for both self and other conditions. Furthermore, the significant four-way ∆*s* × treatment (OXT vs. Placebo) × context (gain vs. loss) × recipient (self vs. other) interaction effect (β=0.125±0.053, *P*=0.018 95% CI=[0.022, 0.228]) in the regression analysis, followed by significant simple effects (In the OXT treatment: ∆*s* × recipient effect in the gain context: *F*_1.45_ = 7.622, *P* < 0.008, *η*^2^ = 0.145; ∆*s* × recipient effect in the loss context: *F*_1.45_ = 7.966, *P* 0.007, *η*^2^ = 0.150, suggested that under OXT treatment, participants showed a greater sensitivity toward ∆*s* (see asterisks in the OXT condition in panel B) in the other condition than the self-condition, thus restoring the hyperaltruistic behavior in loss context.

As the reviewer suggested, OXT’s effect on hyperaltruism does manifest separately on subjects’ harm sensitivities on self- and other-oriented actions. We followed the reviewer’s suggestions and examined the actual magnitude of the sensitivities to shocks for both the self and other treatments (panel B in the figure above). It’s clear that the administration of OXT (compared to the Placebo treatment, panel B in the figure above) significantly reduced participants’ pain sensitivity (treatment × ∆*s*: β=-0.136±0.029, *P* < 0.001, 95% CI=[-0.192,-0.079]), yet also restored the harm sensitivity patterns in both the gain and loss conditions. These results are included in the supplementary figures (6 & 8) as well as in the main texts.

Recommendations:(1) For Figures 2A-B, it would be great to calculate the correlation separately for gain and loss, as in other figures.

We speculate that the reviewer is referring to Figures 3A & B. Sorry that we did not present the correlations separately for the gain and loss contexts because the correlation between an individual’s IH (instrumental harm), IB (impartial beneficence) and hyperaltruistic preferences was not significantly modulated by the contextual factors. The interaction effects in both Figs. 3A & B and Supplementary Fig.5 (also see Table S1& S2) are as following: Study1 valence × IH effect: β=0.016±0.022, *t*_152_=0.726, *P*=0.469; valence × IB effect: β=0.004±0.031, *t*_152_=0.115, *P*=0.908; Study2 placebo condition: valence × IH effect: β=0.018±0.024, *t*_84_=0.030 *P*=0.463; valence × IB effect: β=0.051±0.030, *t*_84_=1.711, *P*=0.702. We have added these statistics to the main text following the reviewer’s suggestions.

(2) "by randomly drawing a shock increment integer ∆s (from 1 to 19) such that [...] did not exceed 20 (𝑆+ {less than or equal to} 20)." I am not sure if a random drawing following a uniform distribution can guarantee S is smaller than 20. More details are needed. Same for the monetary magnitude.

We are sorry for the lack of clarity in the method description. As for the task design, we followed adopted the original design from previous literature (Crockett et al., 2014, 2017). More specifically:

“Specifically, each trial was determined by a combination of the differences of shocks (Δs, ranging from 1 to 19, with increment of 1) and money (Δm, ranging from ¥0.2 to ¥19.8, with increment of ¥0.2) between the two options, resulting in a total of 19×99=1881 pairs of [Δs, Δm]. for each trial. To ensure the trials were suitable for most subjects, we evenly distributed the desired ratio Δm / (Δs + Δm) between 0.01 and 0.99 across 60 trials for each condition. For each trial, we selected the closest [Δs, Δm] pair from the [Δs, Δm] pool to the specific Δm / (Δs + Δm) ratio, which was then used to determine the actual money and shock amounts of two options. The shock amount (*Sless*) for the less painful option was an integer drawn from the discrete uniform distribution [1-19], constraint by *Sless* + ∆s < 20. Similarly, the money amount (*Mless*) for the less painful option was drawn from a discrete uniform distribution [¥0.2 - ¥19.8], with the constraint of M_less_ + ∆m < 20. Once the *Sless*and *Mless* were selected, the shock (*Smore*) and money (*Mmore*) magnitudes for the more painful option were calculated as: *Smore* = S_less_ + ∆s, *Mmore* = M_less_ + ∆m”

We have added these details to the methods section (Lines 520-533).

**Reviewer #2:**
(1) The theoretical hypothesis needs to be better justified. There are studies addressing the neurobiological mechanism of hyperaltruistic tendency, which the authors unfortunately skipped entirely.

Also in recommendation #1:

(1) In the Introduction, the authors claim that "the mechanistic account of the hyperaltruistic phenomenon remains unknown". I think this is too broad of a criticism and does not do justice to prior work that does provide some mechanistic account of this phenomenon. In particular, I was surprised that the authors did not mention at all a relevant fMRI study that investigates the neural mechanism underlying hyperaltruistic tendency (Crockett et al., 2017, Nature Neuroscience). There, the researchers found that individual differences in hyperaltruistic tendency in the same type of moral decision-making task is better explained by reduced neural responses to ill-gotten money (Δm in the Other condition) in the brain reward system, rather than heightened neural responses to others' harm. Moreover, such neural response pattern is related to how an immoral choice would be judged (i.e., blamed) by the community. Since the brain reward system is consistently involved in Oxytocin's role in social cognition and decision-making (e.g., Dolen & Malenka, 2014, Biological Psychiatry), it is important to discuss the hypothesis and results of the present research in the context of this literature.

We totally agree with the reviewer that the expression “mechanistic account of the hyperaltruistic phenomenon remains unknown” in our original manuscript can be misleading to the audience. Indeed, we were aware of the major findings in the field and cited all the seminal work of hyperaltruism and its related neural mechanism (Crockett et al., 2014, 2015, 2017). We have changed the texts in the introduction to better reflect this point and added further discussion as to how oxytocin might play a role:

“For example, it was shown that the hyperaltruistic preference modulated neural representations of the profit gained from harming others via the functional connectivity between the lateral prefrontal cortex, a brain area involved in moral norm violation, and profit sensitive brain regions such as the dorsal striatum6.” (Lines 41~45)

“Oxytocin has been shown to play a critical role in social interactions such as maternal attachment, pair bonding, consociate attachment and aggression in a variety of animal models[42,43]. Humans are endowed with higher cognitive and affective capacities and exhibit far more complex social cognitive patterns[44]. ” (Lines 86~90)

(2) There are some important inconsistencies between the preregistration and the actual data collection/analysis, which the authors did not justify.

Also in recommendations:

(4) It is laudable that the authors pre-registered the procedure and key analysis of the Oxytocin study and determined the sample size beforehand. However, in the preregistration, the authors claimed that they would recruit 30 participants for Experiment 1 and 60 for Experiment 2, without justification. In the paper, they described a "prior power analysis", which deviated from their preregistration. It is OK to deviate from preregistration, but this needs to be explicitly mentioned and addressed (why the deviation occurred, why the reported approach was justifiable, etc.).

We sincerely appreciate the reviewer’s thorough assessment of our manuscript. In the more exploratory study 1, we found that the loss decision context effectively diminished subjects’ hyperaltruistic preference. Based on this finding, we pre-registered study 2 and hypothesized that: (1) The administration of OXT may salvage subject’s hyperaltruistic preference in the loss context; (2) The administration of OXT may reduce subjects’ sensitivities towards electric shocks (but not necessarily their moral preference), due to the well-established results relating OXT to enhanced empathy for others (Barchi-Ferreira & Osório, 2021; Radke et al., 2013) and the processing of negative stimuli(Evans et al., 2010; Kirsch et al., 2005; Wu et al., 2020); and (3) The OXT effect might be context specific, depending on the particular combination of valence (gain vs. loss) and shock recipient (self vs. other) (Abu-Akel et al., 2015; Kapetaniou et al., 2021; Ma et al., 2015).

As our results suggested, the administration of OXT indeed restored subjects’ hyperaltruistic preference (confirming hypothesis 1, Figure 4A). Also, OXT decreased subjects’ sensitivities towards electric shocks in both the gain and loss conditions (supplementary Fig. 6 and supplementary Fig. 8), consistent with our second hypothesis. We must admit that our hypothesis 3 was rather vague, since a seminal study clearly demonstrated the context-dependent effect of OXT in human cooperation and conflict depending on the group membership of the subjects (De Dreu et al., 2010, 2020). Although our results partially validated our hypothesis 3 (supplementary Fig. 6), we did not make specific predictions as to the direction and the magnitude of the OXT effect.

The main inconsistency is related to the sample size. When we carried out study 1, we recruited both male and female subjects. After we identified the context effect on the hyperaltruistic preference, we decided to pre-register and perform study 2 (the OXT study). We originally made a rough estimate of 60 male subjects for study 2. While conducting study 2, we also went through the literature of OXT effect on social behavior and realized that the actual subject number around 45 might be enough to detect the main effect of OXT. Therefore, we settled on the number of 46 (study 2) reported in the manuscript. Correspondingly, we increased the subject number in study 1 to the final number of 80 (40 males) to make sure the subject number is enough to detect a small-to-medium effect, as well as to have a fair comparison between study 1 and 2 (roughly equal number of male subjects). It should be noted that although we only reported all the subjects (male & female) results of study 1 in the manuscript, the main results remain very similar if we only focus on the results of male subjects in study 1 (see the figure below). We believe that these results, together with the placebo treatment group results in study 2 (male only), confirmed the validity of our original finding.

**Author response image 2. sa4fig2:** 

**Author response image 3. sa4fig3:** 

We have included additional texts (Lines 447 ~ 452) in the Methods section for the discrepancy between the preregistered and actual sample sizes in the revised manuscript:

“It should be noted that in preregistration we originally planned to recruit 60 male subjects for Study 2 but ended up recruiting 46 male subjects (mean age = years) based on the sample size reported in previous oxytocin studies[57,69]. Additionally, a power analysis suggested that the sample size > 44 should be enough to detect a small to median effect size of oxytocin (Cohen’s d=0.24, α=0.05, β=0.8) using a 2 × 2 × 2 within-subject design[76].”

(3) Some of the exploratory analysis seems underpowered (e.g., large multiple regression models with only about 40 participants).

We thank the reviewer’s comments and appreciate the concern that the sample size would be an issue affecting the results reliability in multiple regression analysis.

In Fig. 2, the multiple regression analyses were conducted after we observed a valence-dependent effect on hyperaltruism (Fig. 2A) and the regression was constructed accordingly:

*Choice ~ ∆s *context*recipient + ∆m *context*recipient+(1+ ∆s *context*recipient + ∆s*context*recipient | subject)*

Where ∆s and ∆m indicate the shock level and monetary reward difference between the more and loss painful options, context as the monetary valence (gain vs. loss) and recipient as the identity of the shock recipient (self vs. other).

Since we have 240 trials for each subject and a total of 80 subjects in Study 1, we believe that this is a reasonable regression analysis to perform.

In Fig. 3, the multiple regression analyses were indeed exploratory. More specifically, we ran 3 multiple linear regressions:

*hyperaltruism~EC*context+IH*context+IB*context*

*Relative harm sensitivity~ EC*context+IH*context+IB*context*

*Relative money sensitivity~ EC*context+IH*context+IB*context*

Where Hyperaltruism is defined as *κother* - *κself*, Relative harm sensitivity as *otherβ∆s* - *selfβ∆s* and Relative monetary sensitivity as *otherβ∆m* - *selfβ∆m*. EC (empathic concern), IH (instrumental harm) and IB (impartial beneficence) were subjects’ scores from corresponding questionnaires.

For the first regression, we tested whether EC, IH and IB scores were related to hyperaltruism and it should be noted that this was tested on 80 subjects (Study 1). After we identified the effect of IH on hyperaltruism, we ran the following two regressions. The reason we still included IB and EC as predictors in these two regression analyses was to remove potential confounds caused by EC and IB since previous research indicated that IB, IH and EC could be correlated (Kahane et al., 2018).

In study 2, we performed the following regression analyses again to validate our results (Placebo treatment in study 2 should have similar results as found in study 1).

*Relative harm sensitivity~ EC*context+IH*context+IB*context*

*Relative money sensitivity~ EC*context+IH*context+IB*context*

Again, we added IB and EC only to control for the nuance effects by the covariates. As indicated in Fig. 5 C-D, the placebo condition in study 2 replicated our previous findings in study 1 and OXT administration effectively removed the interaction effect between IH and valence (gain vs. loss) on subjects’ relative harm sensitivity.

To more objectively present our data and results, we have changed the texts in the results section and pointed out that the regression analysis:

*hyperaltruism~EC*context+IH*context+IB*context*

was exploratory (Lines 186-192).

“We tested how hyperaltruism was related to both IH and IB across decision contexts using an exploratory multiple regression analysis. Moral preference, defined as *κother* - *κself*, was negatively associated with IH (β=-0.031±0.011, *t*_156_=-2.784, *P* = 0.006) but not with IB (β=0.008±0.016, *t*_156_=0.475, *P*=0.636) across gain and loss contexts, reflecting a general connection between moral preference and IH (Fig. 3A & B).”

(4) Inaccurate conceptualization of utilitarian psychology and the questionnaire used to measure it.

Also in recommendations:

(2) Throughout the paper, the authors placed lots of weight on individual differences in utilitarian psychology and the Oxford Utilitarianism Scale (OUS). I am not sure this is the best individual difference measure in this context. I don't see a conceptual fit between the psychological construct that OUS reflects, and the key psychological processes underlying the behaviors in the present study. As far as I understand it, the conceptual core of utilitarian psychology that OUS captures is the maximization of greater goods. Neither the Instrumental Harm (IH) component nor the Impartial Beneficence (IB) component reflects a tradeoff between the personal interests of the decision-making agent and a moral principle. The IH component is about the endorsement of harming a smaller number of individuals for the benefit of a larger number of individuals. The IB component is about treating self, close others, and distant others equally. However, the behavioral task used in this study is neither about distributing harm between a smaller number of others and a larger number of others nor about benefiting close or distant others. The fact that IH showed some statistical association with the behavioral tendency in the present data set could be due to the conceptual overlap between IH and an individual's tendency to inflict harm (e.g., psychopathy; Table 7 in Kahane et al., 2018, which the authors cited). I urge the authors to justify more why they believe that conceptually OUS is an appropriate individual difference measure in the present study, and if so, interpret their results in a clearer and justifiable manner (taking into account the potential confound of harm tendency/psychopathy).

We thank the reviewer for the thoughtful comment and agree that “IH component is about the endorsement of harming a smaller number of individuals for the benefit of a larger number of individuals. The IB component is about treating self, close others, and distant others equally”. As we mentioned in the previous response to the reviewer, we first ran an exploratory multiple linear regression analysis of hyperaltruistic preference (*κother* - *κself*) against IB and IH in study 1 based on the hypothesis that the reduction of hyperaltruistic preference in the loss condition might be due to (1) subjects’ altered altitudes between IB and hyperaltruistic preference between the gain and loss conditions, and/or (2) the loss condition changed how the moral norm was perceived and therefore affected the correlation between IH and hyperaltruistic preference. As Fig. 3 shows, we did not find a significant IB effect on hyperaltruistic preference (*κother* - *κself*), nor on the relative harm or money sensitivity (supplementary Fig. 3). These results excluded the possibility that subjects with higher IB might treat self and others more equally and therefore show less hyperaltruistic preference. On the other hand, we found a strong correlation between hyperaltruistic preference and IH (Fig. 3A): subjects with higher IH scores showed less hyperaltruistic preference. Since the hyperaltruistic preference (*κother* - *κself*) is a compound variable and we further broke it down to subjects’ relative sensitivity to harm and money (*other β∆s* - *self β∆s* and *other β∆m* - *self β∆m*, respectively). The follow up regression analyses revealed that the correlation between subjects’ relative harm sensitivity and IH was altered by the decision contexts (gain vs. loss, Fig. 3C-D). These results are consistent with our hypothesis that for subjects to engage in the utilitarian calculation, they should first realize that there is a moral dilemma (harming others to make monetary gain in the gain condition). When there is less perceived moral conflict (due to the framing of decision context as avoiding loss in the loss condition), the correlation between subjects’ relative harm sensitivity and IH became insignificant (Fig. 3C). It is worth noting that these results were further replicated in the placebo condition of study 2, further indicating the role of OXT is to affect how the decision context is morally framed.

The reviewer also raised an interesting possibility that the correlation between subject’s behavioral tendency and IH may be confounded by the fact that IH is also correlated with other traits such as psychopathy. Indeed, in the Kahane et al., 2018 paper, the authors showed that IH was associated with subclinical psychopathy in a lay population. Although we only collected and included IB and Empathic concern (EC) scores as control variables and in principle could not rule out the influence of psychopathy, we argue it is unlikely the case. First, psychopaths by definition “only care about their own good” (Kahane et al., 2018). However, subjects in our studies, as well as in previous research, showed greater aversion to harming others (compared to harming themselves) in the gain conditions. This is opposite to the prediction of psychopathy. Even in the loss condition, subjects showed similar levels of aversion to harming others (vs. harming themselves), indicating that our subjects valuated their own and others’ well-being similarly. Second, although there appears to be an association between utilitarian judgement and psychopathy(Glenn et al., 2010; Kahane et al., 2015), the fact that people also possess a form of universal or impartial beneficence in their utilitarian judgements suggest psychopathy alone is not a sufficient variable explaining subjects’ hyperaltruistic behavior.

We have thus rewritten part of the results to clarify our rationale for using the Oxford Utilitarianism Scale (especially the IH and IB) to establish the relationship between moral traits and subjects’ decision preference (Lines 212-215):

“Furthermore, our results are consistent with the claim that profiting from inflicting pains on another person (IH) is inherently deemed immoral1. Hyperaltruistic preference, therefore, is likely to be associated with subjects’ IH dispositions.”

(3) Relatedly, in the Discussion, the authors mentioned "the money-pain trade-off task, similar to the well-known trolley dilemma". I am not sure if this statement is factually accurate because the "well-known trolley dilemma" is about a disinterested third-party weighing between two moral requirements - "greatest good for the greatest number" (utilitarianism) and "do no harm" (Kantian/deontology), not between a moral requirement and one's own monetary interest (which is the focus of the present study). The analogy would be more appropriate if the task required the participants to trade off between, for example, harming one person in exchange for a charitable donation, as a recent study employed (Siegel et al., 2022, A computational account of how individuals resolve the dilemma of dirty money. Scientific reports). I urge the authors to go through their use of "utilitarian/utilitarianism” in the paper and make sure their usage aligns with the definition of the concept and the philosophical implications.

We thank the reviewer for prompting us to think over the difference between our task and the trolley dilemma. Indeed, the trolley dilemma refers to a disinterested third-party’s decision between two moral requirements, namely, the utilitarianism and deontology. In our study, when the shock recipient was “other”, our task could be interpreted as either the decision between “moral norm of no harm (deontology) and one’s self-interest maximization (utilitarian)”, or a decision between “greatest good for both parties (utilitarian) vs. do no harm (deontology)”, though the latter interpretation typically requires differential weighing of own benefits versus the benefits of others(Fehr & Schmidt, 1999; Saez et al., 2015). In fact, it could be argued that the utilitarianism account applies not only to the third party’s well-being, but also to our own well-being, or to “that of those near or dear to us” (Kahane et al., 2018).

We acknowledge that there may lack a direct analogy between our task and the trolley dilemma and therefore have deleted the trolley example in the discussion.

(5) Related to the above point, the sample size of Study 2 was calculated based on the main effect of oxytocin. However, the authors also reported several regression models that seem to me more like exploratory analyses. Their sample size may not be sufficient for these analyses. The authors should: (a) explicitly distinguish between their hypothesis-driven analysis and exploratory analysis; (b) report achieved power of their analysis.

We appreciate the reviewer’s thorough reading of our manuscript. Following the reviewer’s suggestions, we have explicitly stated in the revised manuscript which analyses were exploratory, and which were hypothesis driven. Following the reviewer’s request, we added the achieved power into the main texts (Lines 274-279):

“The effect size (Cohen’s *f*^2^) for this exploratory analysis was calculated to be 0.491 and 0.379 for the placebo and oxytocin conditions, respectively. The post hoc power analysis with a significance level of α = 0.05, 7 regressors (IH, IB, EC, decision context, IH×context, IB×context, and EC×context), and sample size of N = 46 yielded achieved power of 0.910 (placebo treatment) and 0.808 (oxytocin treatment).”

*(6) Do the authors collect reaction times (RT) information? Did the decision context and oxytocin modulate RT? Based on their procedure, it seems that the authors adopted a speeded response task, therefore the RT may reflect some psychological processes independent of choice. It is also possible (and recommended) that the authors use the drift-diffusion model to quantify latent psychological processes underlying moral decision-making. It would be interesting to see if their manipulations have any impact on those latent psychological processes, in addition to explicit choice, which is the endpoint product of the latent psychological processes. There are some examples of applying DDM to this task, which the authors could refer to if they decide to go down this route (Yu et al, 2021, How peer influence shapes value computation in moral decision-making. Cognition.)*

We did collect the RT information for this experiment. As demonstrated in the figure below, participants exhibited significantly longer RT in the loss context compared to the gain context (Study1: the main effect of decision context: *F*_1,79_=20.043, *P* < 0.001, *η*^2^ = 0.202; Study2-placebo: *F*_1.45_=17.177, *P* < 0.001, *η*^2^ = 0.276). In addition to this effect of context, decisions were significantly slower in the other-condition compared to the self-condition

(Study1: the main effect of recipient: *F*_1,79_=4.352, *P* < 0.040, *η*^2^ = 0.052; Study2-placebo: *F*_1,45_=5.601, *P* < 0.022, *η*^2^ = 0.111) which replicates previous research findings (Crockett et al., 2014). However, the differences in response time between recipients was not modulated by decision context (Study1: context × recipient interaction: *F*_1,79_=1.538, *P* < 0.219, *η*^2^ = 0.019; Study2-placebo: *F*_1,45_=2.631, *P* < 0.112, *η*^2^ = 0.055). Additionally, the results in the oxytocin study (study 2) revealed no evidence supporting any effect of oxytocin on reaction time. Neither the main effect (treatment: placebo vs. oxytocin) nor the interaction effect of oxytocin on response time was statistically significant (main effect of OXT treatment: *F*_1,45_=2.380, *P* < 0.230, *η*^2^ = 0.050; treatment × context: *F*_1,45_=2.075, *P* < 0.157*η*^2^ = 0.044; treatment × recipient: *F*_1,45_=0.266, *P* < 0.609, *η*^2^ = 0.006; treatment × context × recipient: *F*_1,45_=2.909, *P* < 0.095, *η*^2^ = 0.061).;

**Author response image 4. sa4fig4:** 

We also agree that it would be interesting to also investigate how the OXT might impact the dynamics of the decision process using a drift-diffusion model (DDM). However, we have already showed in the original manuscript that the OXT increased subjects’ relative harm sensitivities. If a canonical DDM is adopted here, then such an OXT effect is more likely to correspond to the increased drift rate for the relative harm sensitivity, which we feel still aligns with the current framework in general. In future studies, including further manipulations such as time pressure might be a more comprehensive approach to investigate the effect of OXT on DDM related decision variables such as attribute drift rate, initial bias, decision threshold and attribute synchrony.

(7) This is just a personal preference, but I would avoid metaphoric language in a scientific paper (e.g., rescue, salvage, obliterate). Plain, neutral English terms can express the same meaning clearly (e.g., restore, vanish, eliminate).

Again, we thank the reviewer for the suggestion and have since modified the terms.

**Reviewer #3:**
The primary weakness of the paper concerns its framing. Although it purports to be measuring "hyper-altruism" it does not provide evidence to support why any of the behavior being measured is extreme enough to warrant the modifier "hyper" (and indeed throughout I believe the writing tends toward hyperbole, using, e.g., verbs like "obliterate" rather than "reduce"). More seriously, I do not believe that the task constitutes altruism, but rather the decision to engage, or not engage, in instrumental aggression.

We agree with the reviewer (and reviewer # 2) that plain and clear English should be used to describe our results and have since modified those terms.

However, the term “hyperaltruism”, which is the main theme of our study, was originally proposed by a seminal paper (Crockett et al., 2014) and has since been widely adopted in related studies (Crockett et al., 2014, 2015, 2017; Volz et al., 2017; Zhan et al., 2020). The term “hyperaltruism” was introduced to emphasize the difference from altruism (Chen et al., 2024; FeldmanHall et al., 2015; Hu et al., 2021; Hutcherson et al., 2015; Lockwood et al., 2017; Xiong et al., 2020). Hyperaltruism does not indicate extreme altruism. Instead, it simply reflects the fact that “we are more willing to sacrifice gains to spare others from harm than to spare ourselves from harm” (Volz et al., 2017). In other words, altruism refers to people’s unselfish regard for or devotion to the welfare of others, and hyperaltruism concerns subject’s own cost-benefit preference as the reference point and highlights the “additional” altruistic preference when considering other’s welfare. For example, in the altruistic experimental design, altruism is characterized by the degree to which subjects take other people’s welfare into account (left panel). However, in a typical hyperaltruism task design (right panel), hyperaltruistic preference is operationally defined as the difference (*κother* - *κself*) between the degrees to which subjects value others’ harm (*κother*) and their own harm (*κself*).

**Author response image 5. sa4fig5:** 

I found it surprising that a paradigm that entails deciding to hurt or not hurt someone else for personal benefit (whether acquiring a financial gain or avoiding a loss) would be described as measuring "altruism." Deciding to hurt someone for personal benefit is the definition of instrumental aggression. I did not see that in any of the studies was there a possibility of acting to benefit the other participant in any condition. Altruism is not equivalent to refraining from engaging in instrumental aggression. True altruism would be to accept shocks to the self for the other's benefit (e.g., money). The interpretation of this task as assessing instrumental aggression is supported by the fact that only the Instrumental Harm subscale of the OUS was associated with outcomes in the task, but not the Impartial Benevolence subscale. By contrast, the IB subscale is the one more consistently associated with altruism (e.g,. Kahane et al 2018; Amormino at al, 2022) I believe it is important for scientific accuracy for the paper, including the title, to be re-written to reflect what it is testing.

Again, as we mentioned in the previous response, hyperaltruism is a term coined almost a decade ago and has since been widely adopted in the research field. We are afraid that switching such a term would be more likely to cause confusion (instead of clarity) among audience.

Also, from the utilitarian perspective, the gain or loss (or harm) occurred to someone else is aligned on the same dimension and there is no discontinuity between gains and losses. Therefore, taking actions to avoid someone else’s loss can also be viewed as altruistic behavior, similar to choices increasing other’s welfare (Liu et al., 2020).

Relatedly: in the introduction I believe it would be important to discuss the non-symmetry of moral obligations related to help/harm--we have obligations not to harm strangers but no obligation to help strangers. This is another reason I do not think the term "hyper altruism" is a good description for this task--given it is typically viewed as morally obligatory not to harm strangers, choosing not to harm them is not "hyper" altruistic (and again, I do not view it as obviously altruism at all).

We agree with the reviewer’s point that we have the moral obligations not to harm others but no obligation to help strangers (Liu et al., 2020). In fact, this is exactly what we argued in our manuscript: by switching the decision context from gains to losses, subjects were less likely to perceive the decisions as “harming others”. Furthermore, after the administration of OXT, making decisions in both the gain and loss contexts were more perceived by subjects as harming others (Fig. 6A).

The framing of the role of OT also felt incomplete. In introducing the potential relevance of OT to behavior in this task, it is important to pull in evidence from non-human animals on origins of OT as a hormone selected for its role in maternal care and defense (including defensive aggression). The non-human animal literature regarding the effects of OT is on the whole much more robust and definitive than the human literature. The evidence is abundant that OT motivates the defensive care of offspring of all kinds. My read of the present OT findings is that they increase participants' willingness to refrain from shocking strangers even when incurring a loss (that is, in a context where the participant is weighing harm to themselves versus harm to the other). It will be important to explain why OT would be relevant to refraining from instrumental aggression, again, drawing on the non-human animal literature.

We thank the reviewer’s comments and agree that the current understanding of the link between our results of OT with animal literature can be at best described as vague and intriguing. Current literature on OT in animal research suggests that the nucleus accumbens (NAc) oxytocin might play the critical role in social cognition and reinforcing social interactions (Dölen et al., 2013; Dölen & Malenka, 2014; Insel, 2010). Though much insight has already been gained from animal studies, in humans, social interactions can take a variety of different forms, and the consociate recognition can also be rather dynamic. For example, male human participants with self-administered OT showed higher trust and cooperation towards in-group members but more defensive aggression towards out-group members (De Dreu et al., 2010). In another human study, participants administered with OT showed more coordinated out-group attack behavior, suggesting that OT might increase in-group efficiency at the cost of harming out-group members (Zhang et al., 2019). It is worth pointing out that in both experiments, the participant’s group membership was artificially assigned, thus highlighting the context-dependent nature of OT effect in humans.

In our experiment, more complex and higher-level social cognitive processes such as moral framing and moral perception are involved, and OT seems to play an important role in affecting these processes. Therefore, we admit that this study, like the ones mentioned above, is rather hard to find non-human animal counterpart, unfortunately. Instead of relating OT to instrumental aggression, we aimed to provide a parsimonious framework to explain why the “hyperaltruism” disappeared in the loss condition, and, with the OT administration, reappeared in both the gain and loss conditions while also considering the effects of other relevant variables.

We concur with the reviewer’s comments about the importance of animal research and have since added the following paragraph into the revised manuscript (Line 86~90) as well as in the discussion:

“Oxytocin has been shown to play a critical role in social interactions such as maternal attachment, pair bonding, consociate attachment and aggression in a variety of animal models[42,43]. Humans are endowed with higher cognitive and affective capacities and exhibit far more complex social cognitive patterns[44].”

Another important limitation is the use of only male participants in Study 2. This was not an essential exclusion. It should be clear throughout sections of the manuscript that this study's effects can be generalized only to male participants.

We thank the reviewer’s comments. Prior research has shown sex differences in oxytocin’s effects (Fischer-Shofty et al., 2013; Hoge et al., 2014; Lynn et al., 2014; Ma et al., 2016; MacDonald, 2013). Furthermore, with the potential confounds of OT effect due to the menstrual cycles and potential pregnancy in female subjects, most human OT studies have only recruited male subjects (Berends et al., 2019; De Dreu et al., 2010; Fischer-Shofty et al., 2010; Ma et al., 2016; Zhang et al., 2019). We have modified our manuscript to emphasize that study 2 only recruited male subjects.

Recommendations:I believe the authors have provided an interesting and valuable dataset related to the willingness to engage in instrumental aggression - this is not the authors' aim, although also an important aim. Future researchers aiming to build on this paper would benefit from it being framed more accurately.Thus, I believe the paper must be reframed to accurately describe the nature of the task as assessing instrumental aggression. This is also an important goal, as well-designed laboratory models of instrumental aggression are somewhat lacking.

Please see our response above that to have better connections with previous research, we believe that the term hyperaltruism might align better with the main theme for this study.

The research literature on other aggression tasks should also be brought in, as I believe these are more relevant to the present study than research studies on altruism that are primarily donation-type tasks. It should be added to the limitations of how different aggression in a laboratory task such as this one is from real-world immoral forms of aggression. Arguably, aggression in a laboratory task in which all participants are taking part voluntarily under a defined set of rules, and in which aggression constrained by rules is mutual, is similar to aggression in sports, which is not considered immoral. Whether responses in this task would generalize to immoral forms of aggression cannot be determined without linking responses in the task to some real-world outcome.

We agree with the reviewer that “aggression in a lab task …. is similar to aggression in sports”. Our starting point was to investigate the boundary conditions for the hyperaltruism (though we don’t deny that there is an aggression component in hyperaltruism, given the experiment design we used). In other words, the dependent variable we were interested in was the difference between “other” and “self” aggression, not the aggression itself. Our results showed that by switching the decision context from the monetary gain environment to the loss condition, human participants were willing to bear similar amounts of monetary loss to spare others and themselves from harm. That is, hyperaltruism disappeared in the loss condition. We interpreted this result as the loss condition prompted subjects to adopt a different moral framework (help vs. harm, Fig. 6A) and subjects were less influenced by their instrumental harm personality trait due to the change of moral framework (Fig. 3C). In the following study (study 2), we further tested this hypothesis and verified that the administration of OT indeed increased subjects’ perception of the task as harming others for both gain and loss conditions (Fig. 6A), and such moral perception mediated the relationship between subject’s personality traits (instrumental harm) and their relative harm sensitivities (the difference of aggression between the other- and self-conditions). We believe the moral perception framework and that OT directly modulates moral perception better account for subjects’ context-dependent choices than hypothesizing OT’s context-dependent modulation effects on aggression.

The language should also be toned down--the use of phrases like "hyper altruism" (without independent evidence to support that designation) and "obliterate" rather than "reduce" or "eliminate" are overly hyperbolic.

We have changed terms such as “obliterate” and “eliminate” to plain English, as the reviewer suggested.

Reference

Abu-Akel, A., Palgi, S., Klein, E., Decety, J., & Shamay-Tsoory, S. (2015). Oxytocin increases empathy to pain when adopting the other- but not the self-perspective. Social Neuroscience, 10(1), 7–15.

Barchi-Ferreira, A., & Osório, F. (2021). Associations between oxytocin and empathy in humans: A systematic literature review. Psychoneuroendocrinology, 129, 105268.

Berends, Y. R., Tulen, J. H. M., Wierdsma, A. I., van Pelt, J., Feldman, R., Zagoory-Sharon, O., de Rijke, Y. B., Kushner, S. A., & van Marle, H. J. C. (2019). Intranasal administration of oxytocin decreases task-related aggressive responses in healthy young males. Psychoneuroendocrinology, 106, 147–154.

Chen, J., Putkinen, V., Seppälä, K., Hirvonen, J., Ioumpa, K., Gazzola, V., Keysers, C., & Nummenmaa, L. (2024). Endogenous opioid receptor system mediates costly altruism in the human brain. Communications Biology, 7(1), 1–11.

Crockett, M. J., Kurth-Nelson, Z., Siegel, J. Z., Dayan, P., & Dolan, R. J. (2014). Harm to others outweighs harm to self in moral decision making. Proceedings of the National Academy of Sciences of the United States of America, 111(48), 17320–17325.

Crockett, M. J., Siegel, J. Z., Kurth-Nelson, Z., Dayan, P., & Dolan, R. J. (2017). Moral transgressions corrupt neural representations of value. Nature Neuroscience, 20(6), 879–885.

Crockett, M. J., Siegel, J. Z., Kurth-Nelson, Z., Ousdal, O. T., Story, G., Frieband, C., Grosse-Rueskamp, J. M., Dayan, P., & Dolan, R. J. (2015). Dissociable Effects of Serotonin and Dopamine on the Valuation of Harm in Moral Decision Making. Current Biology, 25(14), 1852–1859.

De Dreu, C. K. W., Greer, L. L., Handgraaf, M. J. J., Shalvi, S., Van Kleef, G. A., Baas, M., Ten Velden, F. S., Van Dijk, E., & Feith, S. W. W. (2010). The Neuropeptide Oxytocin Regulates Parochial Altruism in Intergroup Conflict Among Humans. Science, 328(5984), 1408–1411.

De Dreu, C. K. W., Gross, J., Fariña, A., & Ma, Y. (2020). Group Cooperation, Carrying-Capacity Stress, and Intergroup Conflict. Trends in Cognitive Sciences, 24(9), 760–776.

Dölen, G., Darvishzadeh, A., Huang, K. W., & Malenka, R. C. (2013). Social reward requires coordinated activity of nucleus accumbens oxytocin and serotonin. Nature, 501(7466), 179–184.

Dölen, G., & Malenka, R. C. (2014). The Emerging Role of Nucleus Accumbens Oxytocin in Social Cognition. Biological Psychiatry, 76(5), 354–355.

Evans, S., Shergill, S. S., & Averbeck, B. B. (2010). Oxytocin Decreases Aversion to Angry Faces in an Associative Learning Task. Neuropsychopharmacology, 35(13), 2502–2509.

Fehr, E., & Schmidt, K. M. (1999). A Theory of Fairness, Competition, and Cooperation*. The Quarterly Journal of Economics, 114(3), 817–868.

FeldmanHall, O., Dalgleish, T., Evans, D., & Mobbs, D. (2015). Empathic concern drives costly altruism. Neuroimage, 105, 347–356.

Fischer-Shofty, M., Levkovitz, Y., & Shamay-Tsoory, S. G. (2013). Oxytocin facilitates accurate perception of competition in men and kinship in women. Social Cognitive and Affective Neuroscience, 8(3), 313–317.

Fischer-Shofty, M., Shamay-Tsoory, S. G., Harari, H., & Levkovitz, Y. (2010). The effect of intranasal administration of oxytocin on fear recognition. Neuropsychologia, 48(1), 179–184.

Glenn, A. L., Koleva, S., Iyer, R., Graham, J., & Ditto, P. H. (2010). Moral identity in psychopathy. Judgment and Decision Making, 5(7), 497–505.

Hoge, E. A., Anderson, E., Lawson, E. A., Bui, E., Fischer, L. E., Khadge, S. D., Barrett, L. F., & Simon, N. M. (2014). Gender moderates the effect of oxytocin on social judgments. Human Psychopharmacology: Clinical and Experimental, 29(3), 299–304.

Hu, J., Hu, Y., Li, Y., & Zhou, X. (2021). Computational and Neurobiological Substrates of Cost-Benefit Integration in Altruistic Helping Decision. Journal of Neuroscience, 41(15), 3545–3561.

Hutcherson, C. A., Bushong, B., & Rangel, A. (2015). A Neurocomputational Model of Altruistic Choice and Its Implications. Neuron, 87(2), 451–462.

Insel, T. R. (2010). The Challenge of Translation in Social Neuroscience: A Review of Oxytocin, Vasopressin, and Affiliative Behavior. Neuron, 65(6), 768–779.

Kahane, G., Everett, J. A. C., Earp, B. D., Caviola, L., Faber, N. S., Crockett, M. J., & Savulescu, J. (2018). Beyond sacrificial harm: A two-dimensional model of utilitarian psychology. Psychological Review, 125(2), 131–164.

Kahane, G., Everett, J. A. C., Earp, B. D., Farias, M., & Savulescu, J. (2015). ‘Utilitarian’ judgments in sacrificial moral dilemmas do not reflect impartial concern for the greater good. Cognition, 134, 193–209.

Kahneman, D., & Tversky, A. (1979). Prospect Theory: An Analysis of Decision under Risk. Econometrica, 47(2), 263.

Kapetaniou, G. E., Reinhard, M. A., Christian, P., Jobst, A., Tobler, P. N., Padberg, F., & Soutschek, A. (2021). The role of oxytocin in delay of gratification and flexibility in non-social decision making. eLife, 10, e61844.

Kirsch, P., Esslinger, C., Chen, Q., Mier, D., Lis, S., Siddhanti, S., Gruppe, H., Mattay, V. S., Gallhofer, B., & Meyer-Lindenberg, A. (2005). Oxytocin Modulates Neural Circuitry for Social Cognition and Fear in Humans. The Journal of Neuroscience, 25(49), 11489–11493.

Liu, J., Gu, R., Liao, C., Lu, J., Fang, Y., Xu, P., Luo, Y., & Cui, F. (2020). The Neural Mechanism of the Social Framing Effect: Evidence from fMRI and tDCS Studies. The Journal of Neuroscience, 40(18), 3646–3656.

Liu, Y., Li, L., Zheng, L., & Guo, X. (2017). Punish the Perpetrator or Compensate the Victim? Gain vs. Loss Context Modulate Third-Party Altruistic Behaviors. Frontiers in Psychology, 8, 2066.

Lockwood, P. L., Hamonet, M., Zhang, S. H., Ratnavel, A., Salmony, F. U., Husain, M., & Maj, A. (2017). Prosocial apathy for helping others when effort is required. Nature Human Behaviour, 1(7), 131–131.

Losecaat Vermeer, A. B., Boksem, M. A. S., & Sanfey, A. G. (2020). Third-party decision-making under risk as a function of prior gains and losses. Journal of Economic Psychology, 77, 102206.

Lynn, S. K., Hoge, E. A., Fischer, L. E., Barrett, L. F., & Simon, N. M. (2014). Gender differences in oxytocin-associated disruption of decision bias during emotion perception. Psychiatry Research, 219(1), 198–203.

Ma, Y., Liu, Y., Rand, D. G., Heatherton, T. F., & Han, S. (2015). Opposing Oxytocin Effects on Intergroup Cooperative Behavior in Intuitive and Reflective Minds. Neuropsychopharmacology, 40(10), 2379–2387.

Ma, Y., Shamay-Tsoory, S., Han, S., & Zink, C. F. (2016). Oxytocin and Social Adaptation: Insights from Neuroimaging Studies of Healthy and Clinical Populations. Trends in Cognitive Sciences, 20(2), 133–145.

MacDonald, K. S. (2013). Sex, Receptors, and Attachment: A Review of Individual Factors Influencing Response to Oxytocin. Frontiers in Neuroscience, 6. 194.

Markiewicz, Ł., & Czupryna, M. (2018). Cheating: One Common Morality for Gain and Losses, but Two Components of Morality Itself. Journal of Behavior Decision Making. 33(2), 166-179.

Pachur, T., Schulte-Mecklenbeck, M., Murphy, R. O., & Hertwig, R. (2018). Prospect theory reflects selective allocation of attention. Journal of Experimental Psychology: General, 147(2), 147–169.

Radke, S., Roelofs, K., & De Bruijn, E. R. A. (2013). Acting on Anger: Social Anxiety Modulates Approach-Avoidance Tendencies After Oxytocin Administration. Psychological Science, 24(8), 1573–1578.

Saez, I., Zhu, L., Set, E., Kayser, A., & Hsu, M. (2015). Dopamine modulates egalitarian behavior in humans. Current Biology, 25(7), 912–919.

Teoh, Y. Y., Yao, Z., Cunningham, W. A., & Hutcherson, C. A. (2020). Attentional priorities drive effects of time pressure on altruistic choice. Nature Communications, 11(1), 3534.

Tom, S. M., Fox, C. R., Trepel, C., & Poldrack, R. A. (2007). The neural basis of loss aversion in decision-making under risk. Science, 315(5811), 515–518.

Usher, M., & McClelland, J. L. (2004). Loss Aversion and Inhibition in Dynamical Models of Multialternative Choice. Psychological Review, 111(3), 757–769.

Volz, L. J., Welborn, B. L., Gobel, M. S., Gazzaniga, M. S., & Grafton, S. T. (2017). Harm to self outweighs benefit to others in moral decision making. Proceedings of the National Academy of Sciences of the United States of America, 114(30), 7963–7968.

Wu, Q., Mao, J., & Li, J. (2020). Oxytocin alters the effect of payoff but not base rate in emotion perception. Psychoneuroendocrinology, 114, 104608.

Wu, S., Cai, W., & Jin, S. (2018). Gain or non-loss: The message matching effect of regulatory focus on moral judgements of other-orientation lies. International Journal of Psychology, 53(3), 223-227.

Xiong, W., Gao, X., He, Z., Yu, H., Liu, H., & Zhou, X. (2020). Affective evaluation of others’ altruistic decisions under risk and ambiguity. Neuroimage, 218, 116996.

Yechiam, E., & Hochman, G. (2013). Losses as modulators of attention: Review and analysis of the unique effects of losses over gains. Psychological Bulletin, 139(2), 497–518.

Zhan, Y., Xiao, X., Tan, Q., Li, J., Fan, W., Chen, J., & Zhong, Y. (2020). Neural correlations of the influence of self-relevance on moral decision-making involving a trade-off between harm and reward. Psychophysiology, 57(9), e13590.

Zhang, H., Gross, J., De Dreu, C., & Ma, Y. (2019). Oxytocin promotes coordinated out-group attack during intergroup conflict in humans. eLife, 8, e40698.